# Dynamic Gaussians Mesh: Consistent Mesh Reconstruction From Dynamic Scenes

**Isabella Liu, Hao Su[†], Xiaolong Wang[†]**
UC San Diego, [†] Equal advising

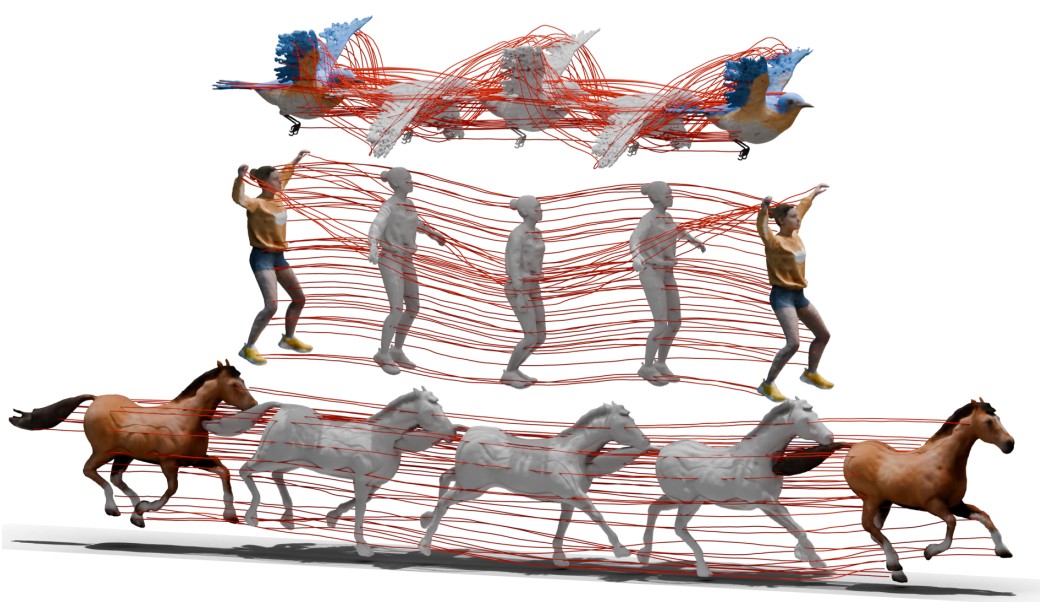

Figure 1: We propose DG-Mesh, a framework that reconstructs high-fidelity time-consistent mesh for dynamic scenes with complex non-rigid deformations. Given dynamic input and the camera parameters, our method reconstructs the high-quality surface and its appearance, as well as the mesh vertice motion across time frames. Our method can reconstruct mesh with flexible topology change as shown above. Additional results can be found on: https://www.liuisabella.com/DG-Mesh.

## Abstract

Modern 3D engines and graphics pipelines require mesh as a memory-efficient representation, which allows efficient rendering, geometry processing, texture editing, and many other downstream operations. However, it is still highly difficult to obtain high-quality mesh in terms of detailed structure and time consistency from dynamic observations. To this end, we introduce Dynamic Gaussians Mesh (DG-Mesh), a framework to reconstruct a high-fidelity and time-consistent mesh from dynamic input. Our work leverages the recent advancement in 3D Gaussian Splatting to construct the mesh sequence with temporal consistency from dynamic observations. Building on top of this representation, DG-Mesh recovers high-quality meshes from the Gaussian points and can track the mesh vertices over time, which enables applications such as texture editing on dynamic objects. We introduce the Gaussian-Mesh Anchoring, which encourages evenly distributed Gaussians, resulting better mesh reconstruction through mesh-guided densification and pruning on the deformed Gaussians. By applying cycle-consistent deformation between the canonical and the deformed space, we can project the anchored Gaussian back to the canonical space and optimize Gaussians across all time frames. During the evaluation on different datasets, DG-Mesh provides significantly better mesh reconstruction and rendering than baselines.

Codes and data are publicly available at https://github.com/Isabella98Liu/DG-Mesh.

# 1 INTRODUCTION

The birds fly, the butterflies flutter, and the flowers sway with the breeze from the wind – our natural world is dynamic, and objects present in the human eyes as an entanglement of structure, motion, and color. A pivotal goal in computer vision is to empower machines with the human-like ability to recover object geometry and motion from visual cues in the environment.

The advent of neural rendering techniques has sparked a surge in research focused on extracting the geometry and motion of dynamic scenes from videos using neural representations. Most studies have concentrated on learning deformable neural radiance fields using volumetric representations (Mildenhall et al., 2021). However, these volumetric models often fall short in terms of memory efficiency and explicit geometric details. The recent introduction of 3D Gaussian Splatting (Kerbl et al., 2023) shifts towards point cloud representations that not only are more memory-efficient but also can provide explicit geometries and superior rendering quality. Further advancements demonstrate how 3D Gaussian Splatting's explicit geometry can effectively represent dynamic scenes by tracking each point's movement across frames (Luiten et al., 2023; Wu et al., 2023a; Yang et al., 2023d).

This paper takes a significant step in the realm of explicit geometry and motion estimation for dynamic scenes. We introduce a method to extract high-fidelity meshes and their motions by tracking vertices over time from videos, as illustrated in Figure 3. Meshes, in contrast to volume, offer a more memory-efficient format. The mapping of correspondences on mesh vertices greatly simplifies texture editing and propagation on target objects. Crucially, the dynamic mesh extracted via our method can be seamlessly integrated into a physical simulator, enabling rapid, physics-based rendering adaptable to diverse materials and lighting conditions.

Our method, Dynamic Gaussians Mesh (DG-Mesh), performs a joint optimization procedure of both 3D Gaussians and the corresponding meshes. Our method not only allows the mesh to have flexible topology changes but also builds the correspondence across meshes over time. Specifically, our method constructs a set of deformable 3D Gaussians by optimizing the 3D Gaussians in a *canonical space* and learning the deformation module for transforming the 3D Gaussians in different time steps for rendering. For each time step, we transform the deformed Gaussians into a mesh in a differentiable manner using a combination of Poisson solver and marching cube algorithm, which is then rendered with a differentiable rasterizer for training. Notably, this fully differentiable pipeline allows the change of mesh topology while maintaining cross-frame consistency implicitly due to the introduction of the canonical space. But how do we find the correspondences given these independent meshes?

We achieve this by decomposing the *mesh-to-mesh* correspondence into two separate correspondences that are easier to acquire: the *mesh-to-points* correspondence that maps the mesh faces to the Gaussian points in each time frame; and the *points-to-canonical-points* correspondence that helps tracking the movement of mesh across time. Our key observation is that the direct training of 3D Gaussians mentioned above will lead to uneven spreading of Gaussians in 3D space (as shown in Figure 2 left). However, this is detrimental for mesh reconstruction performance due to the violation of the Possion solver assumption that points should be evenly distributed; worse still,

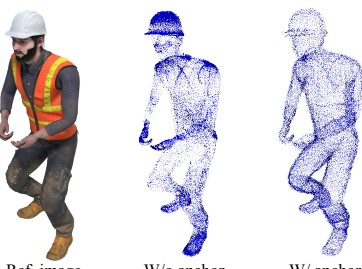

Ref. image    W/o anchor    W/ anchor

Figure 2: The 3D Gaussian centers before and after the Gaussian-Mesh Anchoring.

non-uniform Gaussians make it trickier to track across a long time period. Thus, we propose the **Gaussian-Mesh Anchoring** procedure during training, which yields uniformly distributed Gaussians at each frame. Concretely, first, we encourage deformed 3D Gaussians in each time step to align with the corresponding mesh surface that has uniform face distribution due to the underlying iso-surface algorithm; and second, we allow adding Gaussians *in the world frame at each time step* if certain mesh faces are not covered by existing deformed Gaussians, as well as deduplicting Gaussians for mesh faces that are covered multiple times. We call the new set of Gaussians with 1-1 correspondences to mesh faces as *anchored Gaussians*. As depicted in Figure 2, anchored Gaussians spread more uniformly in 3D.

Finally, we need to update the canonical space to accommodate the added and removed Gaussians in the world frame so that canonical and anchored Gaussians are always consistent. This can be done

by introducing a backward deformation network that computes the deformation from the anchored Gaussians to the canonical Gaussians, by which we can inject new Gaussians in the canonical space or remove unnecessary Gaussians. We define a **Cycle-Consistent Deformation** loss inspired by (Yang et al., 2023b), where the cycle is formed with: a forward deformation from the canonical Gaussians to deformed Gaussians, an anchoring process modifying the deformed Gaussians to the anchored Gaussians, a backward deformation from anchored Gaussians to canonical Gaussians. As a result, we obtain uniformly distributed 3D Gaussians in each time step, and they are aligned with the mesh faces. This makes finding the correspondences between meshes much easier.

We demonstrate high-fidelity mesh reconstruction and tracking in our experiments. A few examples can be found in Figure 1, where the correspondence is shown as the red curves connecting the vertices across time. Even for challenging geometries (e.g., thin structures like bird wings), our method can still perform reconstruction and achieve much better results than previous approaches, which usually fail to reconstruct the geometry or construct an oversized mesh. We also demonstrate downstream applications, such as texture editing with our extracted mesh and correspondence (Figure 26), which is much harder to achieve with non-mesh representations. To the best of our knowledge, this is the first framework that recovers high-fidelity meshes with cross-frame correspondences from dynamic observations.

## 2   RELATED WORK

**View Synthesis in a Dynamic Scene.** The rise of Neural Radiance Fields (NeRFs) (Mildenhall et al., 2021) has largely transformed 3D scene reconstruction and novel view sythesis. Its success has also been extended to dynamic scenes in two directions. One direction is to build dynamic NeRFs (Du et al., 2021; Gao et al., 2021; Li et al., 2021; Xian et al., 2021) which models the motion of the scene by extending the radiance field with an extra time dimension or a latent code. To achieve faster rendering speed and more efficient use of memory, recent works perform volume factorization converting the 4D volume into multiple lower dimension planers or tensors (Cao & Johnson, 2023; Fridovich-Keil et al., 2023; Shao et al., 2023; Xu et al., 2023). Another direction is to construct an additional deformation field that maps point coordinates in different time frames into a canonical space, where large motion and geometry changes can be captured and learned (Park et al., 2021a; Pumarola et al., 2021; Park et al., 2021b; Tretschk et al., 2021; Fang et al., 2022). The recently proposed 3D Gaussians Splatting (Kerbl et al., 2023) extends the volumetric rendering in NeRF by accommodating point clouds. This not only largely improves the speed of neural rendering but also extracts an explicit point cloud structure from images. When applied to dynamic scenes, the same idea of building a deformation field is applied with 3D Gaussians (Yang et al., 2023d; Wu et al., 2023a; Luiten et al., 2023; Li et al., 2024; Yang et al., 2023c). But instead of an implicit field, the deformation field here explicitly tracks the canonical point clouds over time. Inspired by this line of research, we push forward the reconstruction of explicit geometry representation by using meshes with the tracks of the mesh vertices over time from dynamic observations.

**Dynamic Mesh Reconstruction.** Mesh plays an important role in modern 3D engines, which are widely used for simulation, modeling, and rendering applications. Recovering mesh from static scenes has been extensively studied over the years. Mesh template-based methods (Hanocka et al., 2020; Xue et al., 2023; Pan et al., 2019; Wang et al., 2018; Kanazawa et al., 2018; Li et al., 2020) optimize the vertex positions from a template to align with the object surface. They usually assume the template is a sphere or a given prior shape, and learn to deform the template into the desired shape to match with the input images. However, when extended to dynamic scenes, these methods often fail because of the topology change during the deformation. Another line of work extract meshed from the implicit field, which can easily handle topology change during deformation. In particular, the mesh can be extracted by identifying and triangulating the zero level sets with methods like Marching Cubes (Cubes, 1987), Marching Tetrahedra (Treece et al., 1999), and Dual Contouring (Ju et al., 2002). Recently, differentiable mesh extraction methods have been proposed (Liao et al., 2018; Chen et al., 2022; Remelli et al., 2020; Shen et al., 2021; 2023; Wei et al., 2023), directly optimizing mesh through implicit filed. Besides static scenes, several works have been studying extracting the surface geometry of deformable objects from dynamic monocular inputs (Yang et al., 2023a; 2022; Wu et al., 2023b; Yang et al., 2021a;b; Tulsiani et al., 2020; Johnson et al., 2023), which are more closely related to real-world scenarios. Due to the limited information provided by monocular inputs, these methods either rely on category template information or require strong regularization or data modalities to output satisfying results.

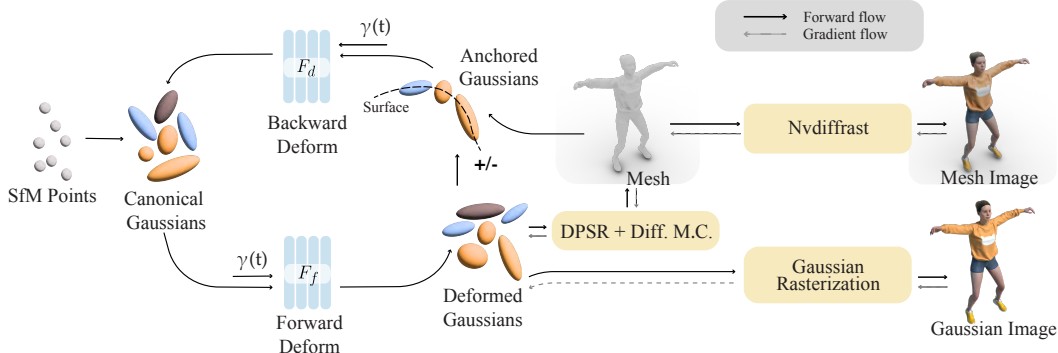

Figure 3: Main pipeline of DG-Mesh. We maintain a set of canonical 3D Gaussians. Under each time step, we transform it into a deformed space. We treat each set of deformed Gaussian points as an oriented point cloud and apply a differentiable Poisson Solver and differentiable Marching Cubes to recover the deformed surface. We propose Gaussian-Mesh Anchoring to adjust the deformed Gaussians to be uniformly aligned with the mesh faces. During anchoring, Gaussian densification and pruning are performed. We use a backward deformation module to project the newly adjusted Gaussian points back to the canonical space.

## 3 METHOD

In this section, we present our framework shown in Figure 3. Given inputs from a dynamic scene, as well as the time label and camera parameters for each input, we reconstruct the dynamic mesh and its appearance, along with the mesh vertex's motion trajectory. The surface correspondence across times benefits many downstream applications, such as dynamic texture editing, which propagates the editing under a single frame to the rest of the frames.

Our method maintains a set of canonical 3D Gaussians and deformation networks to deform the Gaussians into different times (Section 3.1). We use a differentiable Poison Solver and a differentiable Marching Cubes method to reconstruct the oriented 3D Gaussian points into meshes (Section 3.2). Our model allows the *mesh-to-mesh* correspondence. This is constructed by two types of correspondence, including the *mesh-to-point* correspondence under each time step and the *point-to-canonical-point* correspondence across all time frames. We propose the **Gaussian-Mesh Anchoring** to encourage the 1-1 correspondence between the mesh faces and the Gaussians, producing more uniformly distributed *anchored Gaussians*. The **Cycle-Consistent Deformation** maintains the points to canonical points correspondence, which includes a backward deformation network that computes the deformation from the anchored Gaussians to the canonical Gaussians (Section 3.3). Finally, we combine all the training objectives (Section 3.4).

### 3.1 DEFORMABLE 3D GAUSSIAN SPLATTING

**3D Gaussian Splatting.** Recently, 3D Gaussians Splatting (3DGS) (Kerbl et al., 2023) has adopted a novel approach based on point cloud rendering, achieving optimal results in novel viewpoint synthesis and scene modeling. In our work, 3DGS provides a fast modeling to explicit geometry and motion, allowing a high-quality and efficient surface reconstruction from the 3D Gaussian point cloud.

We define each 3D Gaussian $G$ by a position (mean) $\mu$, a full 3D covariance matrix $\Sigma$ centered at point (mean) $\mu$, and the opacity $\alpha$:

$$G(x; \mu, \Sigma) \propto \exp\left(-\frac{1}{2}(x - \mu)^T \Sigma^{-1}(x - \mu)\right)$$

During the $\alpha$-blending process, each Gaussian is multiplied by $\alpha$. To project 3D Gaussians onto image space for rendering, we construct a covariance matrix $\Sigma'$ through $\Sigma' = JW\Sigma W^T J^T$, where $J$ is the Jacobian of the affine approximation of the projective transformation and $W$ is a viewing transformation (world-to-camera transformation matrix). To ensure a positive semi-definite matrix during the optimization process, $\Sigma$ is decomposed into a scaling matrix $S$ and a rotation matrix $R$: $\Sigma = RSS^T R^T$. In our implementation, we directly optimize a 3D vector for scaling $s$ and a quaternion $r$ to represent rotation during the training process. Each 3D Gaussian can now be represented as $G(x; \mu, r, s, \alpha)$.

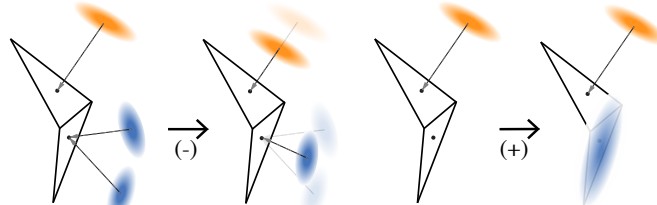

Figure 4: Illustration of our Gaussian-Mesh Anchoring procedure. After obtaining mesh from the DPSR and differentiable Marching Cubes, we adjust the deformed 3D Gaussians to make it more aligned with the mesh's faces. For each Gaussian point, we find its nearest neighbors. If a mesh face is the nearest neighbor to multiple Gaussian points at the same time, we merge these Gaussians and create a new Gaussian from them. If a mesh face is not any Gaussian point's nearest neighbor, we create a new Gaussian on its center.

**Deformable 3D Gaussians.** 3DGS assumes static scenes and its performance drops drastically when reconstructing dynamic scenes. Several works (Wu et al., 2023a; Yang et al., 2023d; Jung et al., 2023; Lin et al., 2023; Zielonka et al., 2023; Li et al., 2024; Yang et al., 2023c) have investigated the way to extend the 3DGS to dynamic reconstruction and an efficient way is to maintain a canonical 3D Gaussains set. We define a transformation function $\mathcal{F}$ which transforms the 3D Gaussian $G(x; \mu, r, s, \alpha)$ from one space to another by:

$$\mathcal{F}(\gamma(x), \gamma(t)) = (\delta x, \delta r, \delta s, \delta \alpha)$$

, where $\gamma$ is the positional encoding function to map $x$ and $t$ to higher dimensional Fourier features (Tancik et al., 2020) to enhance detailed reconstruction:

$$\gamma^k(p) \to (\sin(2^0 \pi p), \cos(2^0 \pi p), ..., \sin(2^k \pi p), \cos(2^k \pi p))$$

The deformed 3D Gaussian in the new space can be represented by:

$$G'(x + \delta x; \mu, r + \delta r, s + \delta s, \alpha + \delta \alpha)$$

### 3.2 MESH RECONSTRUCTION

**Surface Extraction and Rendering Pipeline.** To apply the isosurface algorithms (such as Marching Cubes (Lorensen & Cline, 1998) or DMTet (Shen et al., 2021)) and extract the surface geometry from the point-cloud representation of the 3DGS, an indicator function $\chi(x)$ is required over the 3D grid to describe the geometry. Some work (Tang et al., 2023) query a density grid using 3D Gaussian's opacity, which does not satisfy the gradient condition in the non-boundary space and is computationally inefficient. Instead, we treat 3DGS as an oriented point-cloud and perform a differentiable Poisson solver (DPSR) (Peng et al., 2021) to obtain the $\chi(x)$. We then apply differentiable Marching Cubes (Wei et al., 2023) to generate watertight manifold meshes from the obtained value grid. We additionally optimize vertex color for the mesh and perform differentiable rasterization (Laine et al., 2020) to render the final image.

**Laplacian Regularization.** To better preserve the mesh surface tessellation, we adopt a mesh Laplacian regularization term to penalize local curvature changes. We use a uniformly weighted differential $\delta_i$ of vertex $v_i$ to describe the difference between the position of vertex $v_i$ and the average position of its neighbors. The Laplacian regularization term is the mean square $\delta_i$ for all vertices in the mesh, and $n$ is the total number of vertices of the mesh.

$$\mathcal{L}_{lap} = \frac{1}{n} \sum_{i=0}^{n} ||\delta_i||^2, \delta_i = v_i - \frac{1}{|N_i|} \sum_{k \in N_i} v_k,$$

where $N_i$ is the one-ring neighbor set of vertex $v_i$.

### 3.3 DYNAMIC MESH CORRESPONDENCE

We construct our *mesh-to-mesh* correspondence with two separate correspondences: the *mesh-to-points* correspondence which maps the mesh faces to the Gaussian points in each time frame, and the *point-to-canonical-point* correspondence across all time frames that helps tracking the movement of mesh over time.

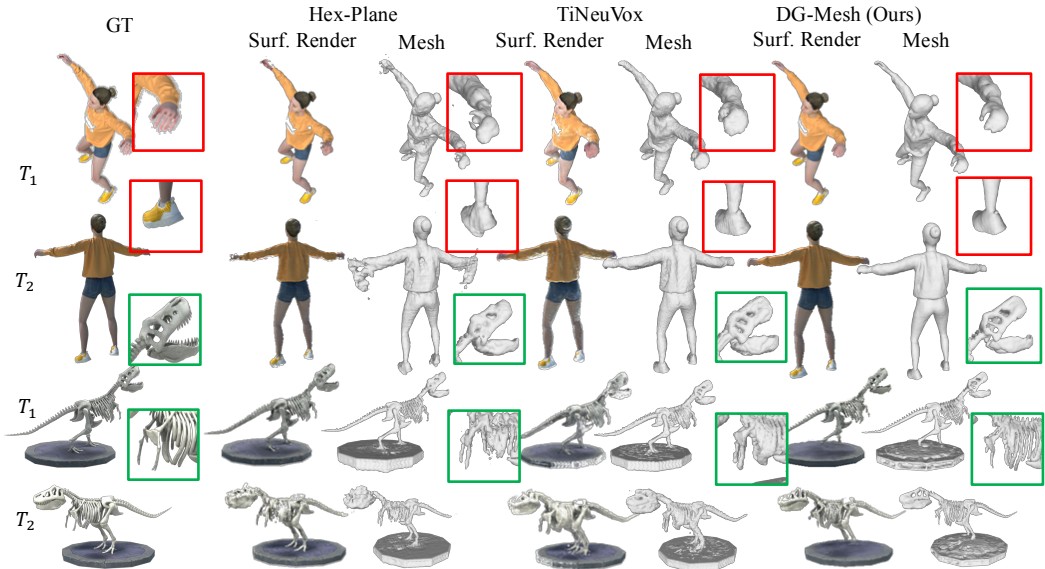

Figure 5: We compare the mesh reconstruction and rendering results of our method with other baselines on the D-NeRF dataset. For each object, we visualize the results under two different time steps and different viewing angles. Our method outperforms others by producing more smooth and detailed structures.

**Gaussian-Mesh Anchoring.** To build the *mesh-to-point* correspondence at each time frame, we introduce the Gaussian-Mesh Anchoring. The original 3D Gaussian densification and deformation mechanism leads to uneven spreading of Gaussians in 3D space (Figure 2), which is undesirable for finding the 1-1 correspondence between mesh and the Gaussians. On the other hand, DPSR discretizes the function values and differential operators in the Poisson equation by assuming the normal vector field $n$ and indicator function $\chi$ are uniformly sampled along each dimension. However, this uniformity is not guaranteed during the above training procedure. To encourage the uniformity of the 3D Gaussian distribution, we use reconstructed mesh to guide the densification and pruning of the 3D Gaussian in the deformed space. Specifically, given a deformed 3D Gaussian set $\{G(x_i)\}_{i=0}^{N}$ and the set of the face centroids from its reconstructed mesh $\{f_j\}_{j=0}^{M}$ and $N \neq M$. For each 3D Gaussian point $x_i$ we find its nearest neighbor in the mesh face set:

$$n_{x_i} = \arg\min_{f} ||x_i - f||, f \in \{f_j\}_{j=0}^{M}$$

During the densification and pruning process, as shown in Figure 4, for face $f_j$ that is the nearest neighbor to multiple $K$ Gaussian points, we prune the original $K$ Gaussians and create a new Gaussian by averaging the properties from the $K$ Gaussians:

$$G' = \frac{1}{K} \sum_{k=0}^{K} G(x_k)$$

On the other hand, for face $f_j$ that is not the nearest neighbor to any Gaussian point, we create a new Gaussian at the face centroid:

$$G' = G(f_j)$$

Besides the above Gaussian density control, we also penalize the distance between Gaussian points and face centroid that has one-to-one correspondence by:

$$\mathcal{L}_{anchor} = \frac{1}{n} \sum_{i=0}^{n} ||x_i - n_{x_i}||^2,$$

where $n$ is the total number of faces that have a one-to-one correspondence to the 3D Gaussian points. A detailed description of the algorithm can be found in Appendix A. The obtained anchored Gaussians $G'$ are applied in the Cycle-Consistent deformation loss $\mathcal{L}_{cycle}$ which will be introduced in the next paragraph.

| Method | Corresp. | Duck | | | | Horse | | | | Bird | | | |
|---|---|---|---|---|---|---|---|---|---|---|---|---|---|
| | | CD ↓ | EMD ↓ | PSNR ↑ | PSNR$_m$ ↑ | CD ↓ | EMD ↓ | PSNR ↑ | PSNR$_m$ ↑ | CD ↓ | EMD ↓ | PSNR ↑ | PSNR$_m$ ↑ |
| D-NeRF | ✗ | 0.934 | 0.073 | 29.785 | 23.019 | 1.685 | 0.280 | 25.474 | 17.381 | 1.532 | 0.163 | 23.848 | 19.573 |
| K-Plane | ✗ | 1.085 | 0.055 | 33.360 | 20.372 | 1.480 | 0.239 | 28.111 | 21.629 | 0.742 | 0.131 | 23.722 | 19.559 |
| HexPlane | ✗ | 2.161 | 0.090 | 32.108 | 27.945 | 1.750 | 0.199 | 26.779 | 22.395 | 4.158 | 0.178 | 22.189 | 20.595 |
| TiNeuVox-B | ✗ | 0.969 | 0.059 | 34.326 | 22.073 | 1.918 | 0.246 | 28.161 | 18.156 | 8.264 | 0.215 | 25.546 | 19.844 |
| 4D-GS | ✗ | 1.134 | 0.111 | 37.127 | - | 1.500 | 0.272 | 29.185 | - | 2.311 | 0.187 | 23.834 | - |
| Deformable-GS | ✗ | 2.366 | 0.115 | 34.187 | - | 1.510 | 0.217 | 30.280 | - | 1.358 | 0.141 | 25.095 | - |
| 4DGS | ✗ | 5.895 | 0.169 | 28.608 | - | 0.895 | 0.340 | 31.138 | - | 20.426 | 0.277 | 24.568 | - |
| SC-GS | ✗ | 1.306 | 0.097 | 40.825 | - | 0.897 | 0.177 | 38.402 | - | 0.897 | 0.166 | 32.575 | - |
| DG-Mesh | ✔ | 0.782 | 0.047 | 28.120 | 32.757 | 0.297 | 0.164 | 23.437 | 30.865 | 0.510 | 0.125 | 24.362 | 28.085 |

| Method | Corresp. | Beagle | | | | Torus2sphere | | | | Girlwalk | | | |
|---|---|---|---|---|---|---|---|---|---|---|---|---|---|
| | | CD ↓ | EMD ↓ | PSNR ↑ | PSNR$_m$ ↑ | CD ↓ | EMD ↓ | PSNR ↑ | PSNR$_m$ ↑ | CD ↓ | EMD ↓ | PSNR ↑ | PSNR$_m$ ↑ |
| D-NeRF | ✗ | 1.001 | 0.149 | 34.470 | 24.446 | 1.760 | 0.250 | 24.227 | 13.562 | 0.601 | 0.190 | 28.632 | 21.146 |
| K-Plane | ✗ | 0.810 | 0.122 | 38.329 | 24.613 | 1.793 | 0.161 | 31.215 | 15.706 | 0.495 | 0.173 | 32.116 | 23.008 |
| HexPlane | ✗ | 0.870 | 0.115 | 38.034 | 29.970 | 2.190 | 0.190 | 29.714 | 22.350 | 0.597 | 0.155 | 31.771 | 24.214 |
| TiNeuVox-B | ✗ | 0.874 | 0.129 | 38.972 | 25.773 | 2.115 | 0.203 | 28.756 | 14.985 | 0.568 | 0.184 | 32.806 | 20.207 |
| 4D-GS | ✗ | 0.644 | 0.106 | 42.995 | - | 2.188 | 0.261 | 28.329 | - | 0.596 | 0.315 | 33.430 | - |
| Deformable-GS | ✗ | 1.154 | 0.161 | 42.530 | - | 2.210 | 0.248 | 28.274 | - | 1.103 | 0.183 | 34.157 | - |
| 4DGS | ✗ | 6.121 | 0.198 | 35.036 | - | 1.523 | 0.226 | 28.533 | - | 1.815 | 0.237 | 35.588 | - |
| SC-GS | ✗ | 3.359 | 0.147 | 41.658 | - | 2.045 | 0.225 | 32.946 | - | 0.623 | 0.203 | 42.615 | - |
| DG-Mesh | ✔ | 0.623 | 0.114 | 29.170 | 33.572 | 1.572 | 0.177 | 20.531 | 25.703 | 0.398 | 0.151 | 25.847 | 33.026 |

Table 1: Mesh reconstruction results of our method compared with other baselines. We measure the reconstructed mesh's: Chamfer Distant (CD) and Earth Mover Distance (EMD) with the ground truth mesh. The unit for CD is $10^{-2}$ for Torus2sphere and $10^{-3}$ for the rest of the scenes. We also measure the volume rendering quality PSNR of the baselines as well as the PSNR$_m$ of their mesh rendering results. We use ▮, ▮, and ▮ to indicate the best, the second best and the third results. Overall, our method produces higher quality meshes among all metrics.

**Cycle-Consistent Deformation.** Deformable 3D Gaussians provide explicit motion and correspondence across times because of their point-cloud representation. During Gaussian-Mesh Anchoring, we align the reconstructed mesh faces and the 3D Gaussian points in each time frame and perform mesh-guided densification and pruning. To update the canonical space to accommodate the anchored Gaussians and build the *point-to-canonical-point* correspondence, we encourage the cycle consistency between the canonical and deformed spaces to ensure that the anchoring performed under the deformed space can be applied back to the canonical space. We define a forward transformation function $\mathcal{F}_f$ and a backward transformation function $\mathcal{F}_b$. Given a canonical 3D Gaussian $G$ and its new form $G'$ under deformed space, we have:

$$\mathcal{F}_f(G, t) = -\mathcal{F}_b(G', t)$$

We use the $L_1$ loss between $\mathcal{F}_f$ and $-\mathcal{F}_d$ as our cycle consistent loss $\mathcal{L}_{cycle}$. This cycle-consistent deformation allows the anchored Gaussian to be deformed back to the canonical space. Specifically, $G'$ will be the anchored Gaussians after the Gaussian-Mesh Anchoring, as introduced previously. This cycle-consistency constraint will encourage adjustments to the canonical Gaussians. This process is visualized on the left side of Figure 3, where the green network is the forward deformation and the blue network is the backward deformation.

### 3.4 Combining All Training Objectives

We use rendering loss from both mesh rasterization ("Mesh Image" in Figure 3) and the 3DGS ("Gaussian Image" in Figure 3) to optimize the mesh geometry and appearance. Given the ground truth image $I_{gt}$ and the Gaussian splatted image $I_{gs}$. The image loss term of 3DGS can be described as:

$$\mathcal{L}_{gs} = (1 - \lambda_{ssim}) \cdot ||I_{gs} - I_{gt}|| + \lambda_{ssim} \cdot \mathcal{L}_{ssim}(I_{gs}, I_{gt})$$

Same for the mesh image loss $\mathcal{L}_{mesh}$. We also compute the $L_1$ loss on the rasterized mask $\mathcal{L}_{mask}$.

The Laplacian loss term $\mathcal{L}_{lap}$ in Section 3.2 helps preserve the mesh surface tessellation and produces smoother surface. The anchoring loss term $\mathcal{L}_{anchor}$ and the cycle-consistency loss term $\mathcal{L}_{cycle}$ described in Section 3.3 help the optimization of cross-frame mesh-to-mesh correspondence. The final loss term can be described as:

$$\mathcal{L} = \mathcal{L}_{gs} + \mathcal{L}_{mesh} + \mathcal{L}_{mask} \\ + \mathcal{L}_{lap} + \mathcal{L}_{anchor} + \mathcal{L}_{cycle}$$

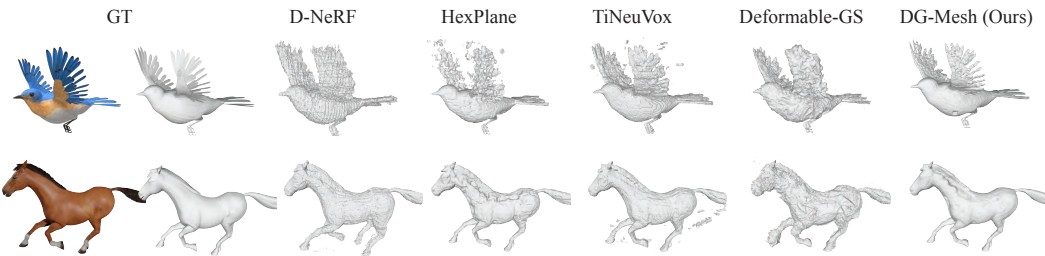

Figure 6: We present the mesh reconstruction results of our method and other baselines on the DG-Mesh dataset. Our method delivers better mesh quality and geometry. For objects like the flying bird, other methods struggle to recover the thin wings, while ours successfully reconstructs them.

## 4 EXPERIMENTS

We introduce the dataset we evaluate on and our implementation details in Section 4.1 and Section 4.2. In Section 4.3, we evaluate our method compared with other baselines. Regarding the baselines, we primarily compare with the following works: D-NeRF (Pumarola et al., 2021), K-Plane (Fridovich-Keil et al., 2023), HexPlane (Cao & Johnson, 2023), TiNeuVox (Fang et al., 2022), 4D-GS (Wu et al., 2023a), Deformable-GS (Yang et al., 2023d), 4DGS (Yang et al., 2023c), and SC-GS (Huang et al., 2024). Among these baselines, D-NeRF (Pumarola et al., 2021) and TiNeu-Vox (Fang et al., 2022) learn a deformation field to transform the dynamic scene into the canonical space, while other methods like HexPlane (Cao & Johnson, 2023) and K-Plane (Fridovich-Keil et al., 2023) utilize volume factorization to learn a compact 4D feature volume. 4D-GS (Wu et al., 2023a), Deformable-GS (Yang et al., 2023d) 4DGS (Yang et al., 2023c), and SC-GS (Huang et al., 2024) use 3D Gaussians as their representations and model their dynamics over time. Note that most of these baselines do not model scene geometry explicitly. To compare the mesh quality, we extract mesh from either an implicit density field or an occupancy field queried from the 3D Gaussians. In Section 4.4, we studied several regularization factors in our method and provided quantitative results of how each factor affects the final performance.

### 4.1 DATASETS

We evaluate our method on the D-NeRF synthetic dataset and provide the visualization and comparison with other baselines in Figure 5. Since the D-NeRF (Pumarola et al., 2021) dataset does not provide mesh ground truth information, we rendered a synthetic dataset containing six dynamic scenes to compare the mesh reconstruction quality quantitatively. Each scene includes 200 frames of a moving object with the ground truth camera parameters and images, as well as the ground truth mesh under each time frame. Camera views are evenly distributed around the target objects on the sphere or the upper sphere. The reconstruction visualization is provided in Figure 6, and full evaluation visualizations can be found in the Appendix L.

For real data evaluation, we run our method on the Nerfies dataset (Park et al., 2021a) , the Dycheck's dataset (Gao et al., 2022) and the Unbiased4D dataset (Johnson et al., 2023), all of which contains monocular videos of everyday deformable objects captured using handheld cameras. To demonstrate the adaptability of DG-Mesh to different dynamic capturing setups, we also test our method on the NeuralActor (Liu et al., 2021) dataset, which features multi-view videos of moving humans. Additionally, we captured several dynamic scenes using a smartphone.

### 4.2 IMPLEMENTATION DETAILS

We utilize the implementation from 3D Gaussian Splatting (Kerbl et al., 2023) for differential Gaussian rasterization. Our model was trained for a total of 50,000 iterations on a single RTX 3090Ti. To better initialize deformable 3D Gaussians, we first trained the canonical Gaussians for $3k$ iterations while keeping the forward and backward deformation network fixed. This helped to retain relatively stable positions and shapes of 3D Gaussians under the canonical space. After $5k$ iterations, we introduce the DPSR and differentiable Marching Cubes to extract the mesh geometry from the Gaussian points. We perform Gaussian-Mesh Anchoring every 100 iteration during training. For the mesh rasterization, we use Nvdiffrast. In order to query the vertex color of the deformed mesh, we project the location of the deformed vertex back to the canonical space and query a time-dependent appearance module to obtain the color. We found optimizing the vertex color in canonical space produces better results compared with querying color under different deformed spaces. The final

| Method | Lego | | | Bouncingballs | | | Jumpingjacks | | | Hook | | |
|---|---|---|---|---|---|---|---|---|---|---|---|---|
| | PSNR$_m$ ↑ | SSIM ↑ | LPIPS ↓ | PSNR$_m$ ↑ | SSIM ↑ | LPIPS ↓ | PSNR$_m$ ↑ | SSIM ↑ | LPIPS ↓ | PSNR$_m$ ↑ | SSIM ↑ | LPIPS ↓ |
| D-NeRF | 20.384 | 0.818 | 0.137 | 23.398 | 0.899 | 0.157 | 22.255 | 0.914 | 0.103 | 20.300 | 0.889 | 0.108 |
| K-Plane | 19.523 | 0.828 | 0.127 | 23.307 | 0.935 | 0.109 | 25.240 | 0.937 | 0.068 | 22.503 | 0.900 | 0.094 |
| HexPlane | 22.872 | 0.904 | 0.072 | 25.389 | 0.957 | 0.069 | 27.078 | 0.954 | 0.052 | 24.513 | 0.929 | 0.070 |
| TiNeuVox-B | 21.927 | 0.843 | 0.126 | 24.819 | 0.947 | 0.101 | 23.621 | 0.932 | 0.075 | 21.429 | 0.908 | 0.085 |
| DG-Mesh | 21.289 | 0.838 | 0.159 | 29.145 | 0.969 | 0.099 | 31.769 | 0.977 | 0.045 | 27.884 | 0.954 | 0.074 |

| Method | Mutant | | | Standup | | | Trex | | | Hellwarrior | | |
|---|---|---|---|---|---|---|---|---|---|---|---|---|
| | PSNR$_m$ ↑ | SSIM ↑ | LPIPS ↓ | PSNR$_m$ ↑ | SSIM ↑ | LPIPS ↓ | PSNR$_m$ ↑ | SSIM ↑ | LPIPS ↓ | PSNR$_m$ ↑ | SSIM ↑ | LPIPS ↓ |
| D-NeRF | 21.070 | 0.906 | 0.077 | 23.380 | 0.925 | 0.069 | 22.594 | 0.908 | 0.085 | 18.907 | 0.877 | 0.129 |
| K-Plane | 23.226 | 0.923 | 0.064 | 25.778 | 0.946 | 0.048 | 23.093 | 0.921 | 0.075 | 18.073 | 0.881 | 0.123 |
| HexPlane | 26.811 | 0.953 | 0.045 | 27.931 | 0.965 | 0.035 | 26.629 | 0.953 | 0.046 | 21.250 | 0.917 | 0.094 |
| TiNeuVox-B | 22.967 | 0.925 | 0.064 | 24.263 | 0.941 | 0.051 | 24.219 | 0.927 | 0.070 | 18.657 | 0.883 | 0.118 |
| DG-Mesh | 30.400 | 0.968 | 0.055 | 30.208 | 0.974 | 0.051 | 28.951 | 0.959 | 0.065 | 25.460 | 0.959 | 0.084 |

Table 2: We measure the mesh rendering PSNR, SSIM and LPIPS score of the baselines and our method on the D-NeRF dataset. Our method has the highest surface rendering quality compared with all the other baselines.

supervision comes from the rendering loss from both the splatted Gaussian images and the rendered mesh images. More implementation details can be found in Appendix B.

## 4.3 RESULTS AND COMPARISONS

For the D-NeRF dataset, we present our numerical results and comparisons in Table 2 and mesh visualization in Figure 5. We compare the mesh rendering PSNR, SSIM, and LPIPS scores for each object in the dataset. Since the D-NeRF dataset does not provide ground truth mesh, we do not numerically evaluate our performance of the mesh quality. In Figure 5, we highlight the part where our method outperforms the other baselines and recovers the highly detailed structures. For the DG-Mesh dataset with mesh ground truth, we provide the numerical results and comparisons in Table 1 and visualizations in Figure 6. Regarding evaluation metrics of mesh quality, we chose two metrics to measure our mesh reconstruction's accuracy: the Chamfer Distance (CD) and the Earth Mover Distance (EMD) to measure the displacement between the reconstructed mesh and the ground truth. As shown in Table 1, our method achieves lower CD and EMD scores compared with other baselines, indicating our method reconstructs the highest quality mesh. In Figure 6, we show the visualization of our reconstructed mesh compared to other baselines in our dataset. In challenging areas, such as thin structures like the wings of the bird, our method still recovers high-quality geometry and surfaces, while other methods produce floaters or oversized meshes. For real evaluation, we provide the quantitative results on the Nerfies dataset in Table 5 and qualitative results in Appendix G. Our method achieves the highest mesh query speed while delivering rendering quality comparable to the baseline. Performance on other monocular datasets is detailed in the supplementary materials, including the Unbiased4D dataset in Appendix J, the Dycheck dataset in Appendix H, and the self-captured iPhone dataset in Appendix I. Additionally, we present qualitative evaluations on the NeuralActor dataset in Appendix K, demonstrating the broader applicability of DG-Mesh to multiview setups.

## 4.4 ABLATION STUDY

**Laplacian Smoothness Regularizer.** We study the effect of laplacian regularizer and its different performance under different loss weights in this part. Laplacian regularization term calculates the average square Euclidean difference between the vertex and its one-ring neighbors, which discourages deformation that will introduce large surface curvature change. We measure the final mesh rendering quality (PSRN, SSIM, LPIPS) under different laplacian loss weights. As shown in Table 3, when $w_{lap} = 1000$, the network produces the highest mesh rendering quality. With appropriate laplacian regularization, our method recovers a smoother surface.

**Gaussian-Mesh Anchoring.** We study the effect of our proposed Gaussian-Mesh Anchoring procedure. Since the original densification and pruning mechanism of 3DGS tends to split more Gaussian in hard areas where the geometry and appearance are more complicated. The original Gaussian center distribution is not even on the object's surface. As discussed in Section 3.3, DPSR assumes uniformly sampled points among the space. An uneven point cloud on the surface will produce the wrong distance field and harm the optimization process. Our Gaussian-Mesh Anchoring performs densification and pruning that will result in a more evenly distributed Gaussian point and rely on

|  | CD $\downarrow$ | EMD $\downarrow$ | PSNR$_m$ $\uparrow$ | SSIM$_m$ $\uparrow$ | LPIPS$_m$ $\downarrow$ |
|---|---|---|---|---|---|
| $w_{lap} = 0$ | 0.972 | 0.089 | 31.947 | 0.971 | 0.079 |
| $w_{lap} = 100$ | 1.239 | 0.093 | 32.178 | 0.969 | 0.069 |
| $w_{lap} = 1000$ | **0.762** | 0.073 | **33.890** | **0.982** | **0.061** |
| $w_{lap} = 2000$ | 0.894 | **0.069** | 32.891 | 0.981 | 0.072 |

Table 3: Results of mesh reconstruction and rendering quality with different Laplacian regularizer ratios. When $w_{lap} = 1000$ our method produces the best results. During the training process, we set the Laplacian ratio to be 1000.

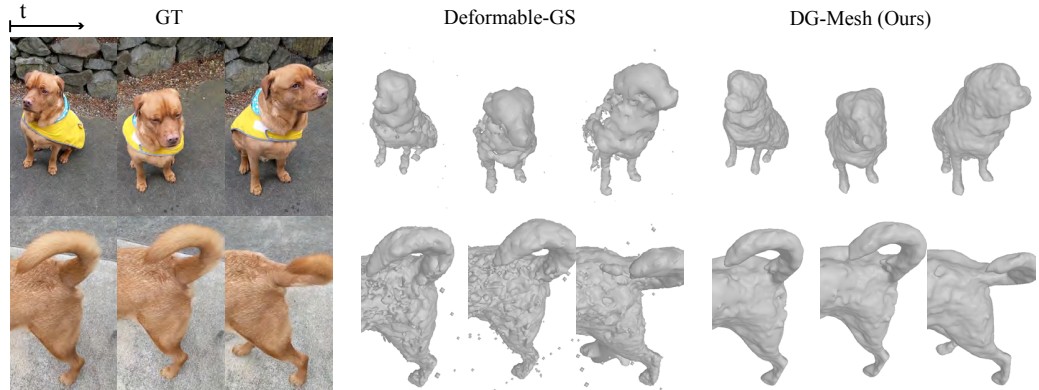

Figure 7: We provide the reconstructed mesh visualization and mesh rendering image of our method on the Nerfies real-world datasets. Our method recovers high-quality and consistent mesh surfaces across multiple time frames, showcasing its robustness and effectiveness in handling dynamic scenes.

the backward deformation network to project it back to the canonical space. As shown in Figure 2, after applying Gaussian-Mesh Anchoring, the Gaussian centers are more evenly distributed on the object's surface. We provide quantitative results of how the Gaussian-Mesh Anchoring procedure allows the network to produce better geometry in Table 4. Without Gaussian-Mesh Anchoring, the reconstructed CD and EMD are both higher.

| Method | CD $\downarrow$ | EMD $\downarrow$ | PSNR$_m$ $\uparrow$ |
|---|---|---|---|
| w./o. anchor | 0.685 | 0.146 | 31.283 |
| w./ anchor | **0.516** | **0.128** | **32.734** |

Table 4: Ablation study on the Gaussian-Mesh Anchoring. The anchoring allows mesh to be better aligned with the ground truth surface.

| Method | PSNR $\uparrow$ | PSNR$_m$ $\uparrow$ | FPS$_m$ $\uparrow$ |
|---|---|---|---|
| Deformable-GS | 28.310 | - | 0.10 |
| DG-Mesh | - | 27.238 | **10.9** |

Table 5: Real scene evaluation on the Nerfies dataset. Our method maintains high mesh querying FPS and comparable image rendering quality to the baseline.

## 5 CONCLUSION

We introduce Dynamic Gaussians Mesh (DG-Mesh), a framework to reconstruct high-fidelity mesh and perform motion tracking from dynamic observations. Leveraging the recent advancements in 3D Gaussian Splatting, DG-Mesh learns a deformable 3D Gaussians set and builds a novel time-consistent mesh on top of it. DG-Mesh outperforms previous methods in reconstructing intricate structures like bird wings and horse legs, which are typically difficult to capture. Moreover, DG-Mesh's ability to provide cross-frame correspondences enables various downstream applications, such as texture editing on dynamic objects, making it a versatile tool with great potential in graphics applications. We will release our code upon paper publication.

**Limitations.** DG-Mesh is currently applied mainly on the foreground objects in a video. To make DG-Mesh work with real-world videos, we require segmentation of the foreground object across the videos. While DG-Mesh handles both the correspondence and topology changes, obtaining correspondence becomes fundamentally challenging in the presence of significant topology changes. For instance, if an additional object or object part appears in the video over time, establishing correspondence for the new object (or object part) with the first frame becomes difficult. This challenge stems from the nature of the problem itself, rather than a limitation of our method.

## ACKNOWLEDGMENTS

This work was supported, in part, by the NSF CAREER Awards (IIS-2240014, IIS-2240160) and by the Qualcomm Embodied AI Fund.

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

APPENDIX TABLE OF CONTENTS

# A  GAUSSIAN-MESH ANCHORING DETAILS

We employ the Gaussian-Mesh Anchoring procedure to guide the Gaussians into a more uniform distribution across the object's surface. This uniformity enhances the smoothness of the mesh reconstruction obtained from DPSR and DiffMC. A more detailed algorithm is provided in Algorithm 1.

The algorithm begins by finding the nearest mesh face for each Gaussian point. For every mesh face, the following steps are taken:

- **If a mesh face is linked to multiple Gaussian points**, we merge these Gaussians to create a new, single Gaussian. This merging helps in reducing redundancy and promotes uniform distribution.

- **If a mesh face is linked to only one specific Gaussian**, we adjust the position of that Gaussian by moving it towards the centroid of the face. This adjustment ensures that the Gaussian is more accurately anchored to the mesh surface.

- **If a mesh face is not linked to any Gaussian point**, we create a new Gaussian at the face's centroid. This step fills in any gaps in the Gaussian distribution, ensuring that all areas of the mesh are adequately represented.

By following this procedure, we ensure that the Gaussians are evenly distributed over the mesh surface, which is crucial for achieving high-quality mesh reconstructions. The anchoring process systematically addresses the association between Gaussians and mesh faces, leading to a more accurate and smooth representation of the object's geometry.

---

**Algorithm 1:** Gaussian-Mesh Anchoring

**Input**  : $\mathcal{G}$ - set of Gaussians in canonical space
$\qquad\quad\;\; \mathcal{V}$ - mesh vertices
$\qquad\quad\;\; \mathcal{F}$ - mesh faces
$\qquad\quad\;\; F_f(\cdot, t)$ - forward deformation function
$\qquad\quad\;\; F_b(\cdot, t)$ - backward deformation function
$\qquad\quad\;\; t$ - timestep
$\qquad\quad\;\; r_s$ - search radius

**Output:** $\mathcal{L}_{anchor}$ - anchoring loss
$\qquad\quad\;\; \mathcal{G}$ - updated set of Gaussians in canonical space

$\mathcal{G}_{\text{def}} \leftarrow \{\mathbf{g}_{\text{def}} = F_f(\mathbf{g}, t) \mid \mathbf{g} \in \mathcal{G}\}$;
$\mathcal{L}_{anchor} \leftarrow 0$;
**foreach** $f \in \mathcal{F}$ **do**
$\quad$ $c_f \leftarrow \text{centroid}(f)$ ;$\qquad\qquad\qquad\qquad$ // Compute the centroid for current face
$\quad$ $\mathcal{G}_f \leftarrow \{\|\mathbf{g}_{\text{def}} - \mathbf{c}_f\| \le r_s \mid \mathbf{g}_{\text{def}} \in \mathcal{G}_{\text{def}}\}$ ;$\qquad$ // Find Gaussians within radius $r_s$ of $\mathbf{c}_f$
$\quad$ **if** $|\mathcal{G}_f| > 1$ **then**
$\quad\quad$ $\mathbf{g}_{\text{new}} \leftarrow \text{merge}(\mathcal{G}_f)$;
$\quad\quad$ $\mathbf{g}_{\text{canonical}} \leftarrow \mathcal{F}_b(\mathbf{g}_{\text{new}}, t)$ ;$\qquad$ // Map the merged Gaussian back to the canonical space
$\quad\quad$ $\mathcal{G} \leftarrow \mathcal{G} \setminus \{\mathcal{F}_b(\mathbf{g}_{\text{def}}, t) \mid \mathbf{g}_{\text{def}} \in \mathcal{G}_f\}$ ;$\qquad$ // Remove the original Gaussians
$\quad\quad$ $\mathcal{G} \leftarrow \mathcal{G} \cup \{\mathbf{g}_{\text{canonical}}\}$ ;$\qquad\qquad\qquad$ // Add the new Gaussian
$\quad\quad$ $\mathcal{L}_{\text{anchor}} \leftarrow \mathcal{L}_{\text{anchor}} + \|\mathbf{g}_{\text{canonical}} - \mathbf{c}_f\|$
$\quad$ **else**
$\quad\quad$ **if** $|\mathcal{G}_f| = 1$ **then**
$\quad\quad\quad$ $\mathbf{g}_{\text{canonical}} \leftarrow \mathcal{F}_b(\mathbf{g}_{\text{def}}, t), \quad \mathbf{g}_{\text{def}} \in \mathcal{G}_f$
$\quad\quad\quad$ $\mathcal{L}_{\text{anchor}} \leftarrow \mathcal{L}_{\text{anchor}} + \|\mathbf{g}_{\text{canonical}} - \mathbf{c}_f\|$
$\quad\quad\quad$ $\mathcal{G} \leftarrow \mathcal{G} \cup \{\mathbf{g}_{\text{canonical}}\}$
$\quad\quad$ **else**
$\quad\quad\quad$ $\mathbf{g}_{\text{new}} \leftarrow \mathbf{c}_f$ ;$\qquad\qquad\qquad$ // Create new Gaussian at the face centroid
$\quad\quad\quad$ $\mathbf{g}_{\text{canonical}} \leftarrow \mathcal{F}_b(\mathbf{g}_{\text{new}}, t)$ ;$\qquad\qquad$ // Add the new Gaussian
$\quad\quad$ **end**
$\quad$ **end**
**end**

---

## B    IMPLEMENTATION DETAILS AND NETWORK ARCHITECTURE

The model was trained over 50,000 iterations using a single NVIDIA RTX 3090Ti GPU, which handled the computational demands effectively. To achieve a more effective initialization of the deformable 3D Gaussians, we began by training the canonical Gaussians for 3,000 iterations. During this initial phase, the forward and backward deformation networks were kept fixed and not updated. This approach helped in maintaining relatively stable positions and shapes of the 3D Gaussians within the canonical space, providing a good initialization for subsequent deformation learning.

After reaching 5,000 iterations, we introduced the Differentiable Poisson Surface Reconstruction (DPSR) (Peng et al., 2021) and differentiable Marching Cubes algorithms (Lorensen & Cline, 1998) to extract mesh geometry from the Gaussian points. This integration allowed us to transition smoothly from point-based representations to mesh-based geometry. Throughout the training process, we performed Gaussian-Mesh Anchoring every 100 iterations to ensure consistency between the Gaussian and mesh representations.

For mesh rasterization, we employed Nvdiffrast (Laine et al., 2020), a GPU-accelerated differentiable renderer. To determine the vertex colors of the deformed mesh, we projected the positions of the deformed vertices back into the canonical space and used a time-dependent appearance module to retrieve the color information. We discovered that optimizing the vertex colors within the canonical space yielded better results compared to querying colors in various deformed spaces. This is likely due to the more stable and consistent representation in the canonical space. The final supervision signal for our model came from the rendering losses calculated from both the splatted Gaussian images and the rendered mesh images, allowing the model to learn from both point-based and mesh-based representations.

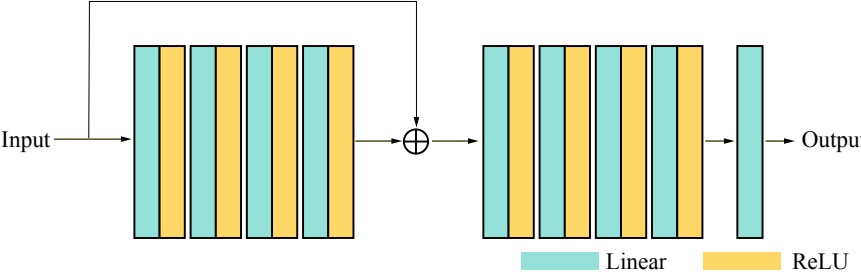

Figure 8: Results comparison on the DG-Mesh dataset. We show the reconstructed mesh and the mesh rendering image. Our method reconstructs better geometry and appearance than other baselines.

Regarding the architecture of the deformation networks, both the forward and backward deformation networks are designed as multi-layer perceptrons (MLPs) with a depth of $D = 8$ layers and hidden layers of dimensionality $W = 256$. The detailed architecture is illustrated in Figure 8. We apply positional encoding functions $\gamma^k(p)$ to map the spatial position $x$ and the temporal label $t$ into higher-dimensional feature spaces. Specifically, we set the encoding parameter $k = 10$ for the positions $x$ of the 3D Gaussians and $k = 6$ for the time labels $t$, enhancing the network's ability to capture high-frequency variations in both space and time. After the final hidden layer, we use a single linear layer without any activation function to produce the predicted offsets for Gaussian position ($\delta x$), scaling ($\delta s$), rotation ($\delta r$), and opacity ($\delta \alpha$). These offsets adjust the Gaussians' parameters based on the input, allowing for dynamic deformation. The appearance network, which is responsible for determining the color of the vertices, shares a similar architectural design with the deformation network.

## C    ABLATIONS STUDY ON THE TEMPLATE-BASED DEFORMATION METHOD

DG-Mesh reconstructs the surface of a mesh across different timesteps by first deforming a set of canonical 3D Gaussians into a deformed space. The surface is then recovered by leveraging the deformed Gaussian points and their corresponding normals. This method allows the model to manage flexible topology changes during deformation, distinguishing it from other approaches that maintain a canonical mesh and learn to deform it over time. We evaluated our method in an extreme scenario where the dynamic scene involves a sphere transforming into a torus. In this case, the genus of the object shifts from 0 to 1. To compare, we ran a baseline experiment in which a network attempted to deform a canonical mesh of a sphere into a torus. The results in Figure 9 indicate that traditional mesh deformation-based methods **struggle to handle topology changes**, whereas our approach effectively overcomes this limitation by using Gaussian points for representation, thereby avoiding the need to manage face connectivity during deformation.

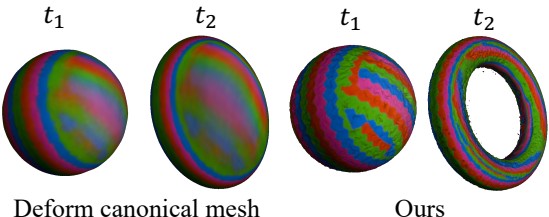

Figure 9: We evaluated our method in an extreme scenario where a sphere deforms into a torus. The traditional mesh deformation-based approach fails in this case because it cannot handle the changes in face connectivity during the deformation process. In contrast, our method successfully reconstructs a high-quality mesh with a completely different topology. This is because we use 3D Gaussian points as our canonical representation, which avoids any issues related to managing mesh face connectivity throughout the deformation.

## D    ABLATIONS STUDY ON FINE-TUNING THE EXTRACTED MESH

**Volume rendering V.S. mesh rendering PSNR.** Previous baselines primarily rely on volume rendering, which offers advantages in view synthesis **but comes with a significant computational cost**. This is because volume rendering integrates color from multiple points along a ray to compute the pixel color, whereas mesh rendering retrieves color from only a single surface point. We compare the inference speed of volume rendering baselines and our mesh rendering in Table 6. Additionally, higher-quality results from volume rendering do not necessarily translate into better geometry or surface reconstruction, as highlighted by the mesh reconstruction comparisons in our paper. To ensure a fairer comparison of rendering PSNR, we provide **the PSNR of mesh rendering after fine-tuning the baselines with a mesh rendering loss** over 20,000 iterations. Even after this fine-tuning, our method still outperforms the baselines in terms of mesh rendering PSNR.

| Methods | Fine-tuned $PSNR_m$ | Training Time | Inference Speed |
|---|---|---|---|
| K-Plane | 24.683 | $\sim 65$ min | 0.9 FPS |
| TiNeuVox | 24.149 | $\sim 70$ min | 0.3 FPS |
| Deformable-GS | 22.519 | $\sim 35$ min | 185 FPS |
| Ours | 29.782 | $\sim 80$ min | 10.9 FPS |

Table 6: We present the fine-tuned mesh rendering results for the baselines, alongside a comparison of training and inference times. Even after fine-tuning the mesh rendering of other baseline methods, our approach still delivers the best results. This highlights the efficiency and effectiveness of our method, both in terms of mesh quality and overall performance.

## E    ABLATION STUDY ON THE APPEARANCE MODULE

We use the geometry generated by DG-Mesh and render the mesh using appearance queried from other baseline methods. The results below demonstrate that, even when the underlying geometry

quality is comparable, DG-Mesh consistently achieves better visual quality compared to other approaches. The surface rendering approach in DG-Mesh enables it to learn appearance directly and precisely at the object's surface. In contrast, other volume rendering-based methods fail to store appearance details accurately at the surface, leading to less precise results.

| Methods | $PSNR_m$ | SSIM |
|---|---|---|
| D-NeRF | 24.169 | 0.905 |
| HexPlane | 29.013 | 0.964 |
| TiNeuVox-B | 28.157 | 0.943 |
| DG-Mesh (Ours) | 29.790 | 0.959 |

Table 7: Comparison of rendering results shows DG-Mesh achieves superior visual quality by accurately learning surface appearance, unlike volume-based methods that lose surface detail.

## F  ABLATION STUDY ON THE ANCHORING FREQUENCY

We evaluate the impact of different anchor intervals and provide their quantitative results below. The mesh geometry quality deteriorates as the anchor intervals increase since larger intervals result in fewer anchoring steps, reducing the alignment between the Gaussians and the surface. Based on the ablation results, we set the anchoring interval to 100, as it represents a balanced trade-off between maintaining geometry quality and achieving optimal rendering performance.

| Anchor Interval | $PSNR_m$ | CD | EMD |
|---|---|---|---|
| 50 | 30.521 | 0.314 | 1.307 |
| 100 | 30.668 | 0.306 | 1.298 |
| 200 | 30.599 | 0.319 | 1.295 |
| 500 | 30.645 | 0.320 | 1.298 |

Table 8: Quantitative results show that increasing anchor intervals deteriorates mesh geometry quality due to fewer anchoring steps, with an interval of 100 providing the best balance between geometry quality and rendering performance. The unit for CD is $10^{-2}$ and the unit for EMD is $10^{-1}$.

## G    More Qualitative Results on the Nerfies Dataset

We evaluate our method on the Nerfies (Park et al., 2021a) dataset, which consists of monocular video sequences capturing everyday moving objects, such as pets. The camera poses throughout the sequences are predominantly forward-facing. To obtain the foreground mask, we utilize an off-the-shelf segmentation tool from (Cheng et al., 2023), and for the camera parameters, we follow the pipeline outlined in (Gao et al., 2022). In Figure 10, we display our mesh reconstruction and rendering results at three different timesteps. Our method consistently recovers high-quality and coherent mesh surfaces over time, demonstrating its effectiveness in dynamic scenes.

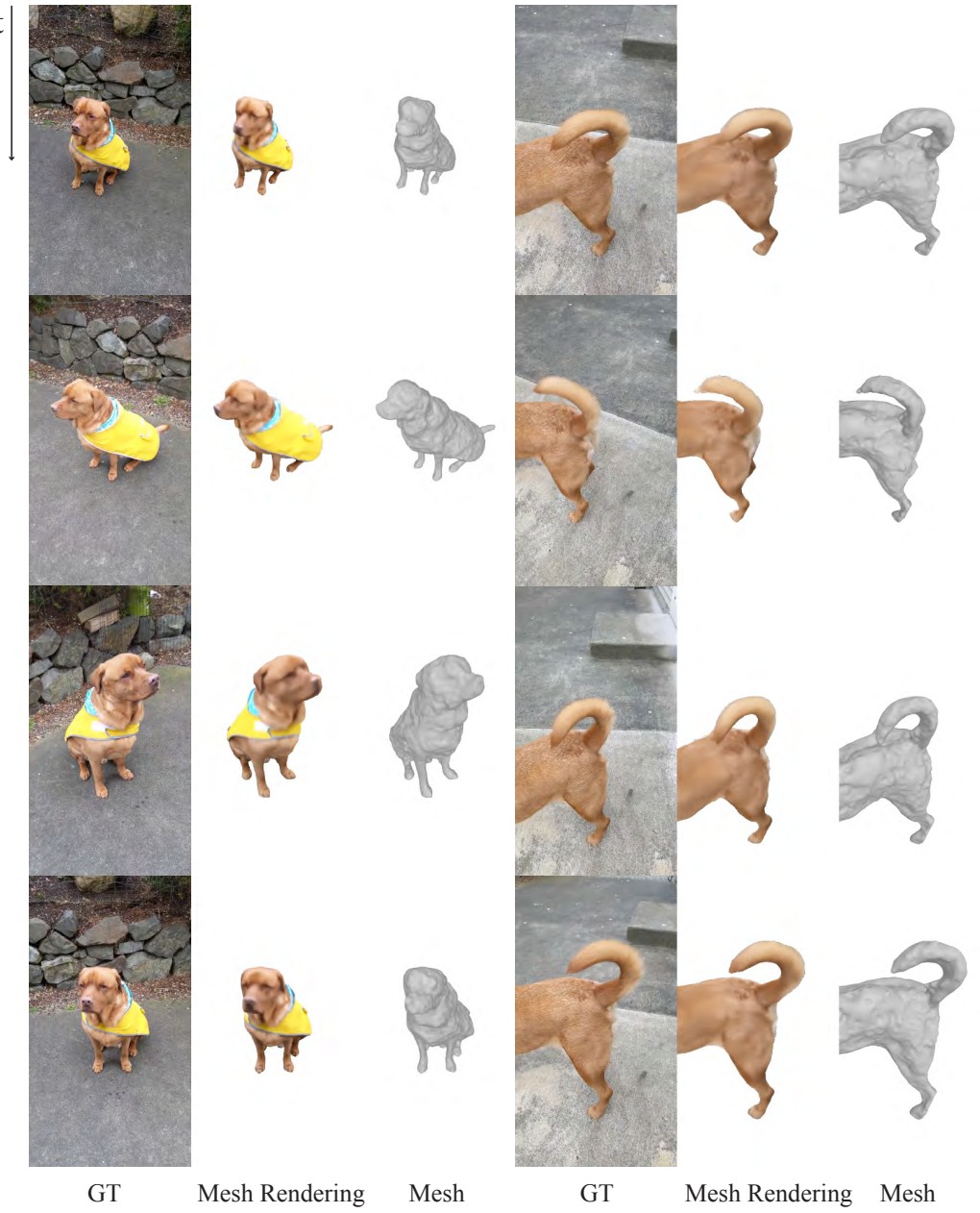

GT        Mesh Rendering      Mesh            GT        Mesh Rendering    Mesh

Figure 10: Mesh reconstruction and rendering results on the Nerfies dataset. Our method recovers high-quality and consistent mesh surfaces across multiple time frames, showcasing its robustness and effectiveness in handling dynamic scenes.

## H    MORE QUALITATIVE RESULTS ON THE DYCHECK DATASET

We evaluate our method on the DyCheck dataset (Gao et al., 2022), which consists of monocular videos captured using smartphones, and compare its mesh reconstruction and rendering quality against Deformable-GS (Yang et al., 2023d). The results show that our method achieves better mesh geometry reconstruction, particularly in scenarios with limited observations from monocular videos. However, we acknowledge that, similar to other dynamic Gaussian-based methods, our approach faces challenges when dealing with extreme object motion or imperfect segmentation.

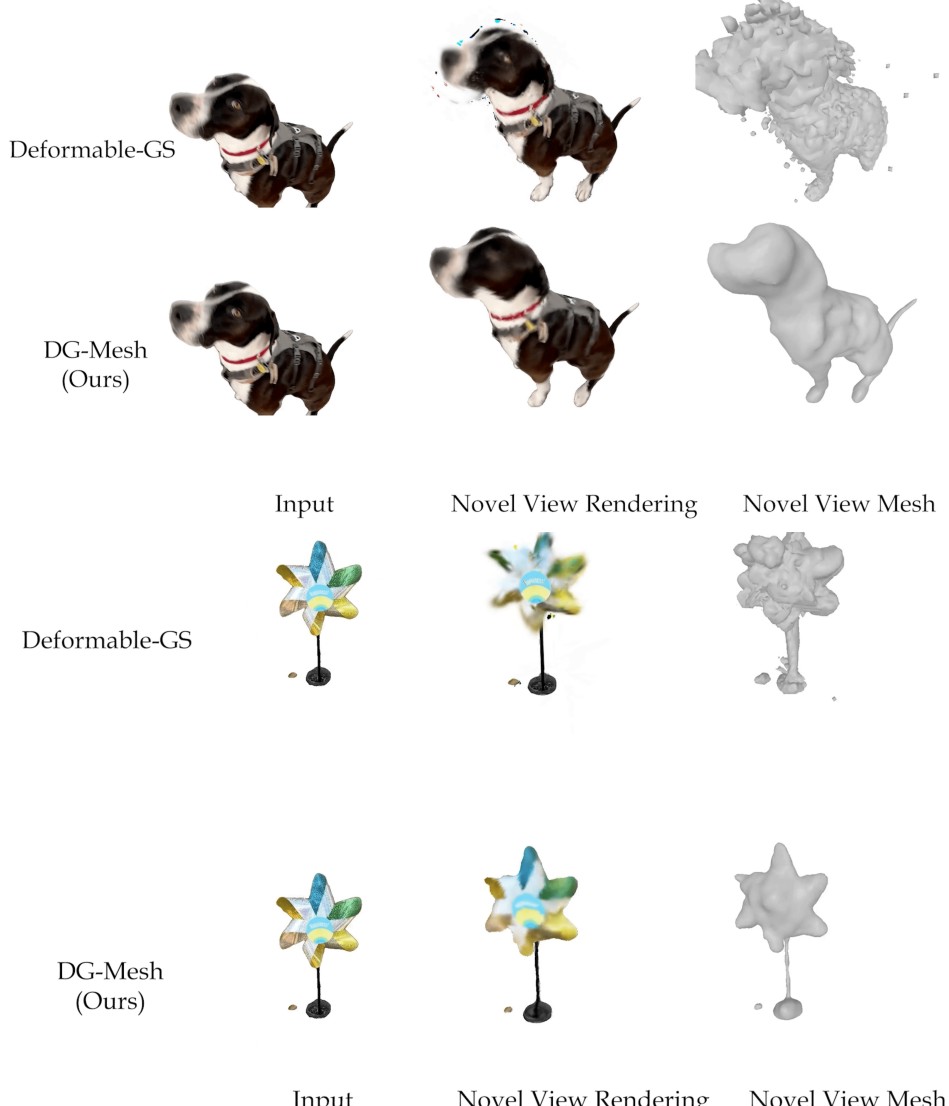

Figure 11: Comparison of mesh reconstruction results with Deformable-GS on the DyCheck dataset. DG-Mesh demonstrates better reconstruction of both mesh geometry and appearance.

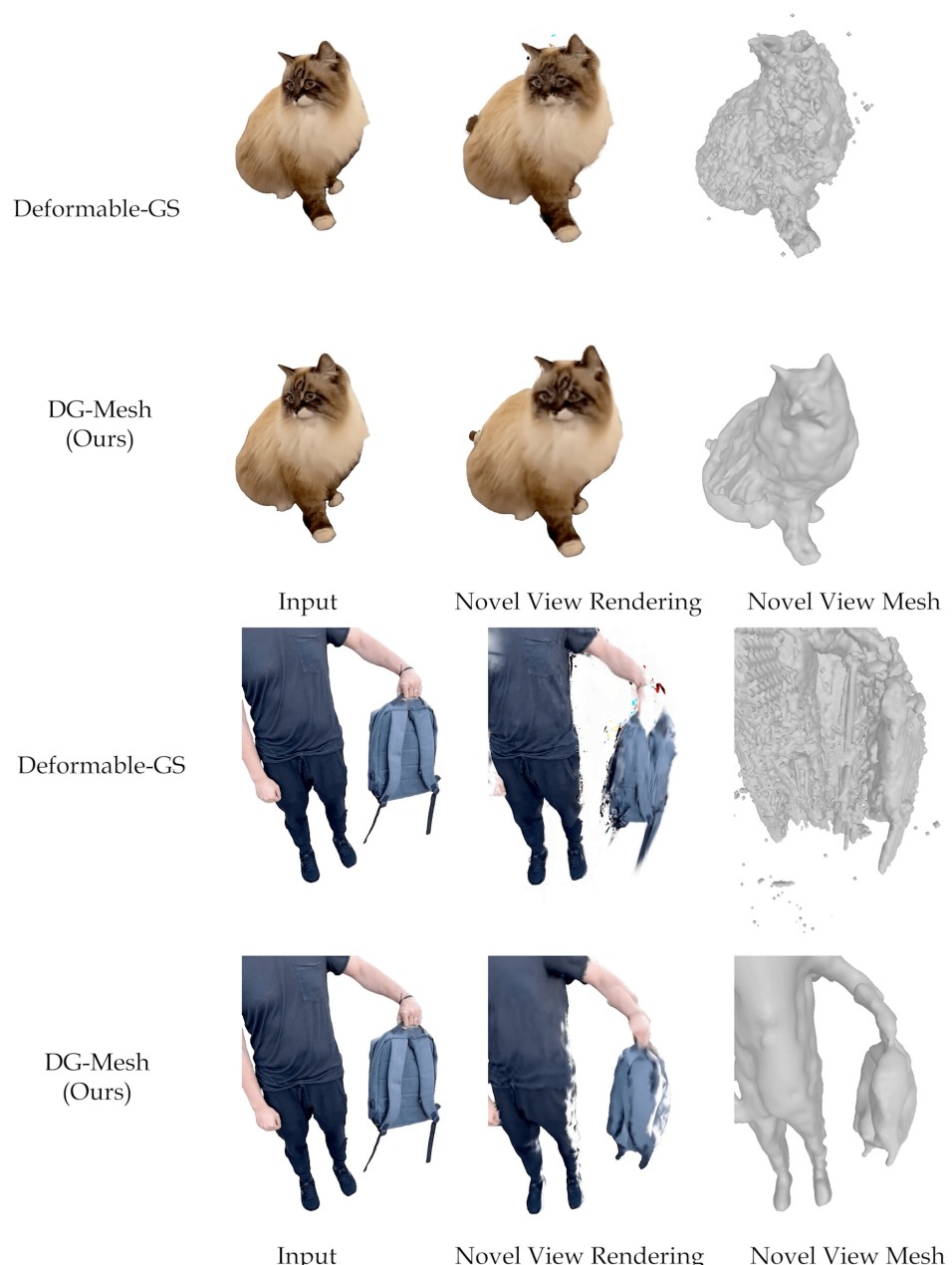

Figure 12: Fail Cases on the DyCheck Dataset: Similar to other dynamic Gaussian-based methods, our approach encounters limitations in scenarios with extreme object motion or imperfect segmentation.

# I    MORE QUALITATIVE RESULTS ON THE SELF-CAPTURED IPHONE DATASET

We captured several casual, everyday videos of moving objects using a smartphone and processed them using a similar data processing pipeline to the one proposed in (Gao et al., 2022). To obtain the data masks, we used an off-the-shelf segmentation tool from (Cheng et al., 2023). The self-captured videos have a frame rate of 60 FPS, with the camera rotating 360 degrees around the object over 8-10 seconds. In Figure 13, we present the mesh reconstruction and rendering results. Our method demonstrates the ability to recover high-quality mesh surfaces from casual smartphone videos, highlighting its practicality and robustness for real-world applications.

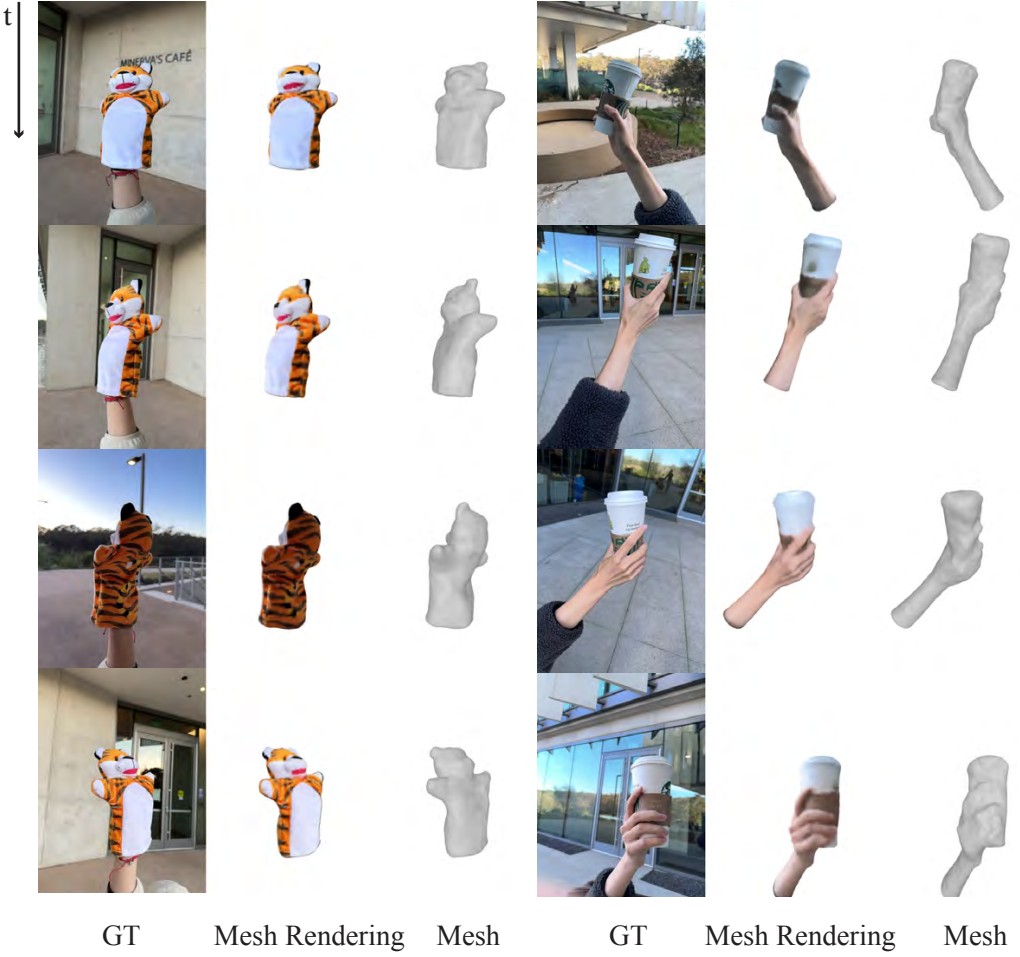

GT    Mesh Rendering    Mesh    GT    Mesh Rendering    Mesh

Figure 13: We present the mesh reconstruction and rendering results on the self-captured iPhone dataset. Our method successfully recovers high-quality and consistent mesh surfaces over multiple time frames, demonstrating its robustness and efficiency in processing real-world dynamic scenes captured in casual smartphone videos.

## J    MORE QUALITATIVE RESULTS ON THE UNBIASED4D DATASET

e evaluate our method on a real video sequence from the Unbiased4D dataset (Johnson et al., 2023) and compare it with the results produced by the method proposed in their paper. The video captures a fast-moving cactus toy, making surface recovery particularly challenging due to the high-speed motion. As shown in Figure 14, our method achieves superior geometry recovery, especially in the hand area of the toy during rapid movement, whereas the Unbiased4D method produces blurrier results and incorrect geometry. Additionally, we provide our mesh reconstruction and rendering results for the sequence at different timeframes in Figure 15, showcasing the robustness of our approach in handling fast-moving objects.

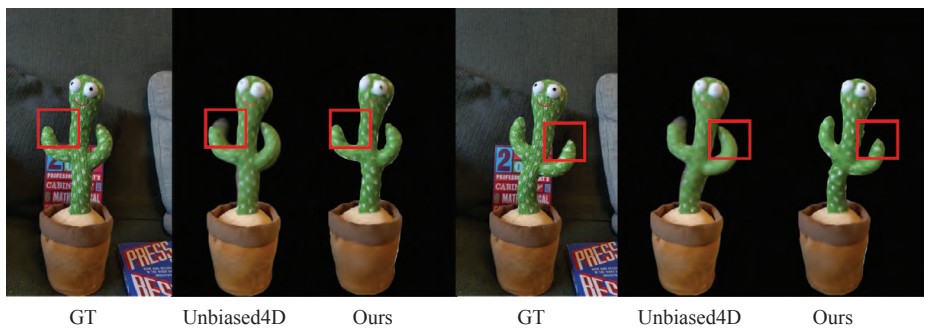

GT          Unbiased4D          Ours          GT          Unbiased4D          Ours

Figure 14: Comparison of the mesh results with the Unbiased4D method. Our approach achieves outstanding geometry recovery, particularly in the hand area of the toy during rapid movement, while the Unbiased4D method results in blurrier outputs and incorrect geometry.

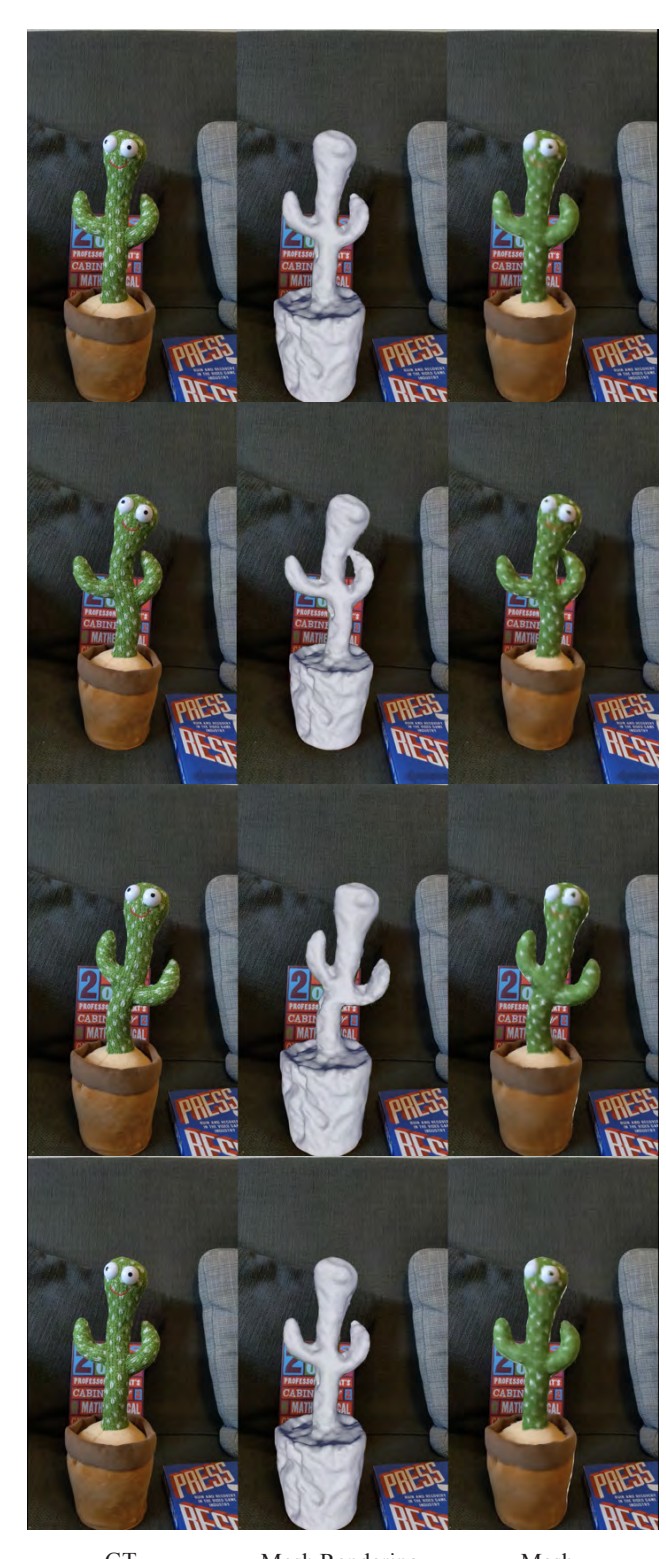

GT                    Mesh Rendering              Mesh

Figure 15: Mesh reconstruction and rendering results on the Unbiased4D dataset. The results indicate the robustness of our approach in handling fast-moving objects.

## K MORE QUALITATIVE RESULTS ON THE NEURALACTOR DATASET

We evaluate our method on the NeuralActor dataset (Liu et al., 2021), a multi-view dynamic dataset featuring videos of moving humans captured from multiple angles. In Figure 16, we present the reconstructed meshes and corresponding mesh rendering results at different time frames. The visualizations demonstrate that, with the benefit of multi-view observations, our method achieves even better surface reconstruction compared to the monocular setup, highlighting its effectiveness in leveraging additional views for more accurate and detailed mesh recovery.

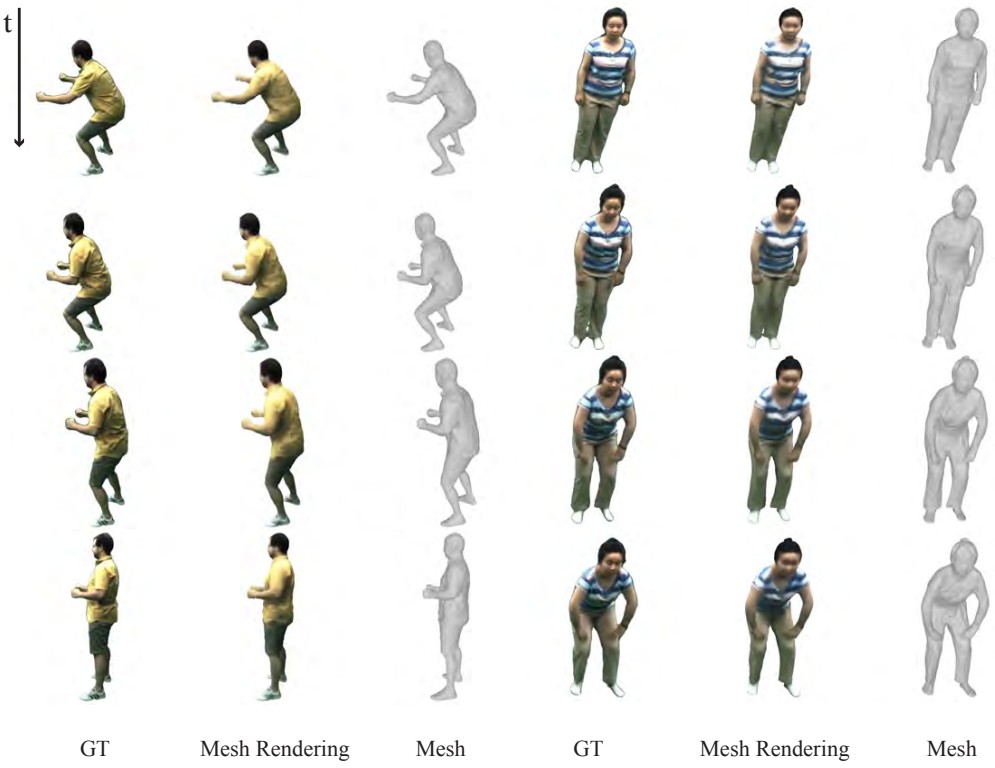

Figure 16: Mesh reconstruction and rendering results on the NeuralActor dataset. The result highlights the effectiveness of our approach in utilizing multiple viewpoints to produce more accurate and detailed mesh recovery in dynamic scenes.

## L   MORE QUALITATIVE RESULTS ON THE D-NERF DATASET

We present the mesh reconstruction results of DG-Mesh across all eight samples from the D-NeRF dataset. For each sample, we provide visualizations of the reconstructed mesh along with the mesh surface rendering results at two different time frames, $t0$ and $t1$. We compare our visual outcomes with those from several baseline methods, including D-NeRF (Pumarola et al., 2021), HexPlane (Cao & Johnson, 2023), K-Plane (Fridovich-Keil et al., 2023), and TiNeuVox (Fang et al., 2022). The results clearly demonstrate that our method achieves superior mesh reconstruction and rendering quality compared to these baselines.

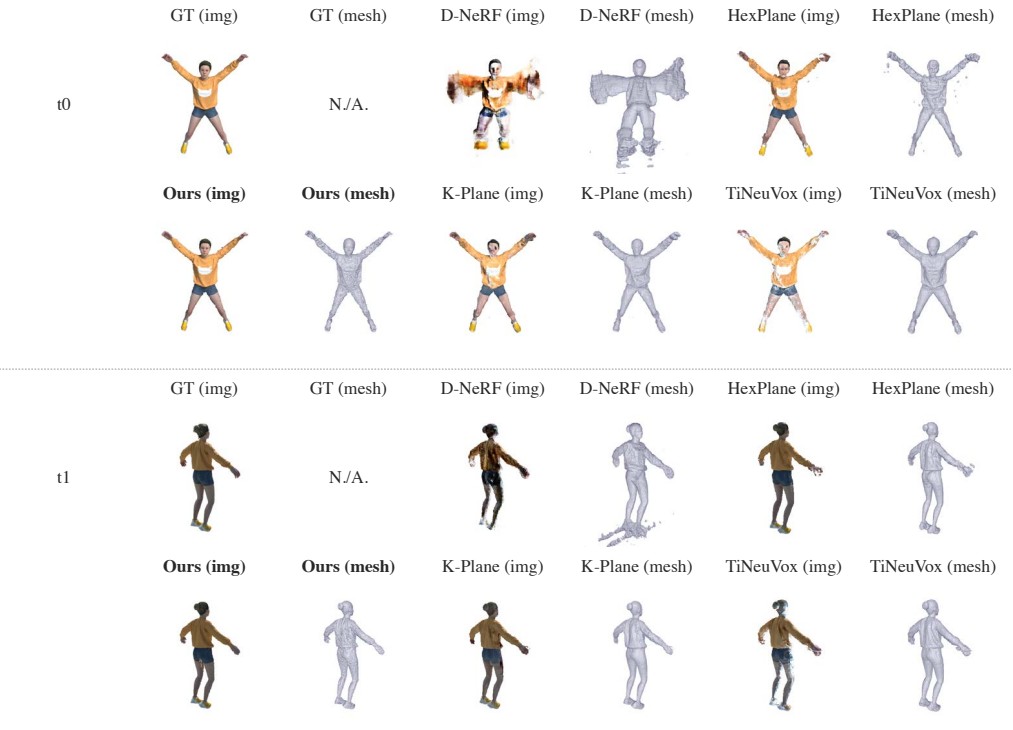

Figure 17: Results comparison on the D-NeRF dataset. We show the reconstructed mesh and the mesh rendering image. Our method reconstructs better geometry and appearance than other baselines.

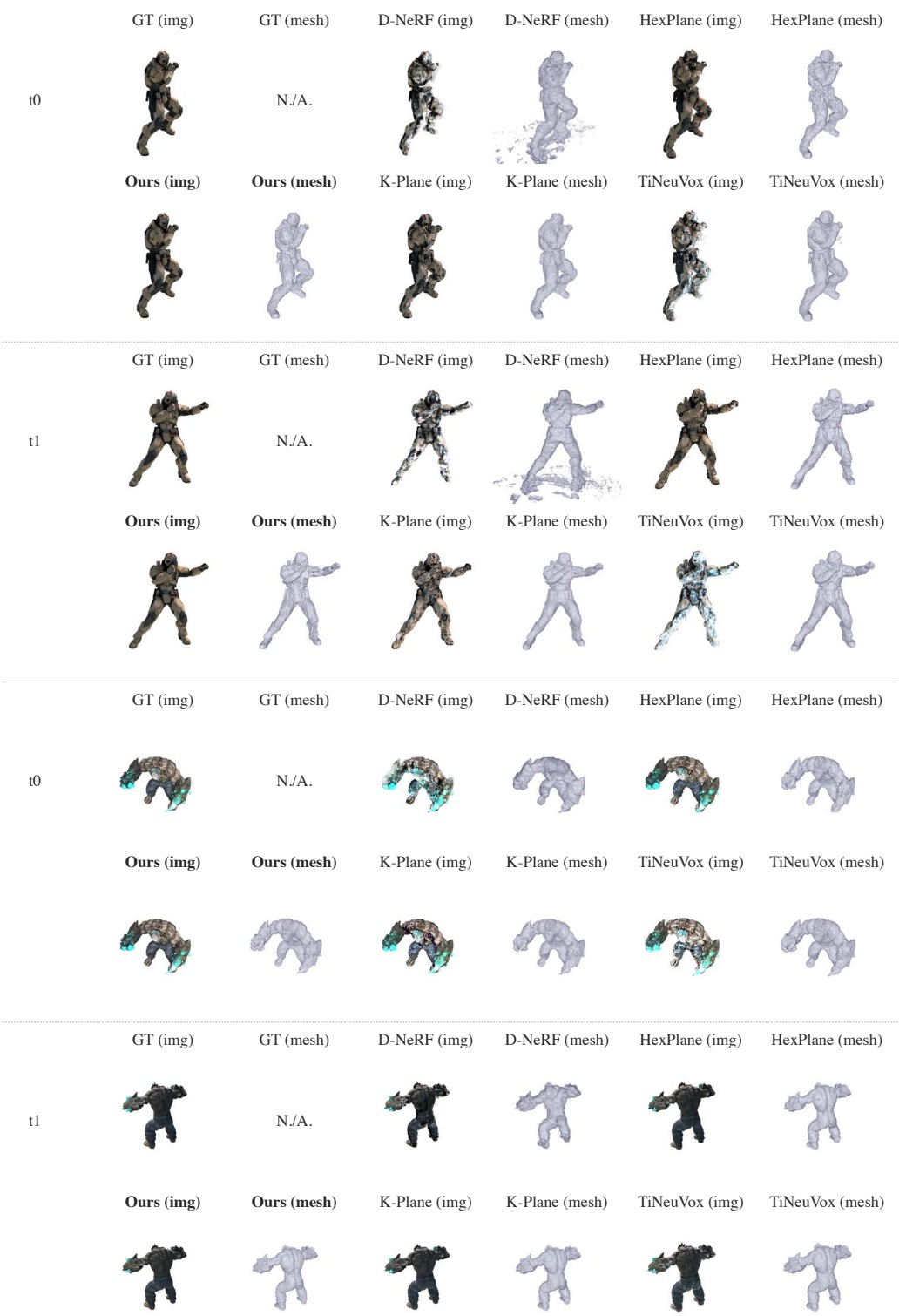

Figure 18: Results comparison on the D-NeRF dataset. We show the reconstructed mesh and the mesh rendering image. Our method reconstructs better geometry and appearance than other baselines.

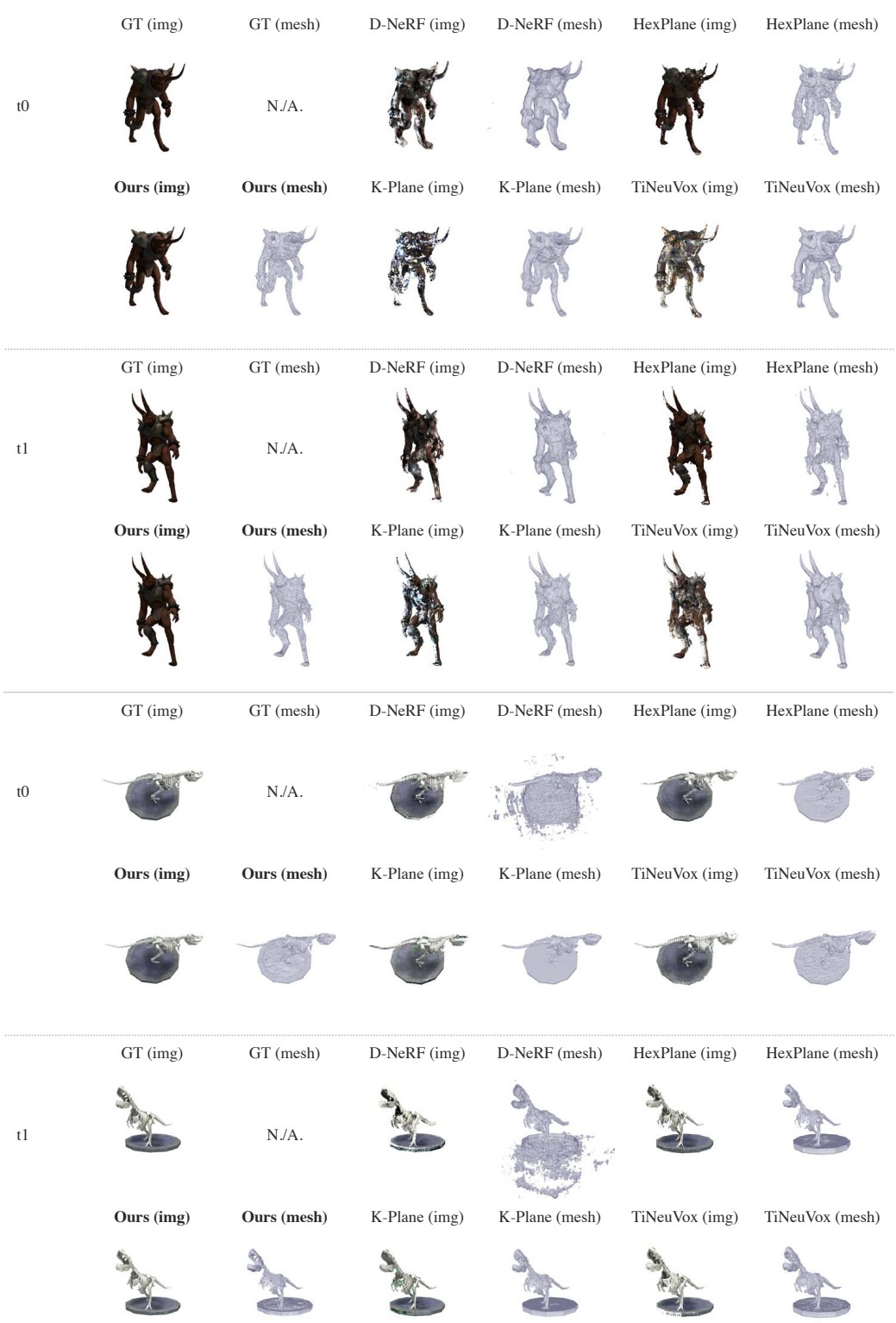

Figure 19: Results comparison on the D-NeRF dataset. We show the reconstructed mesh and the mesh rendering image. Our method reconstructs better geometry and appearance than other baselines.

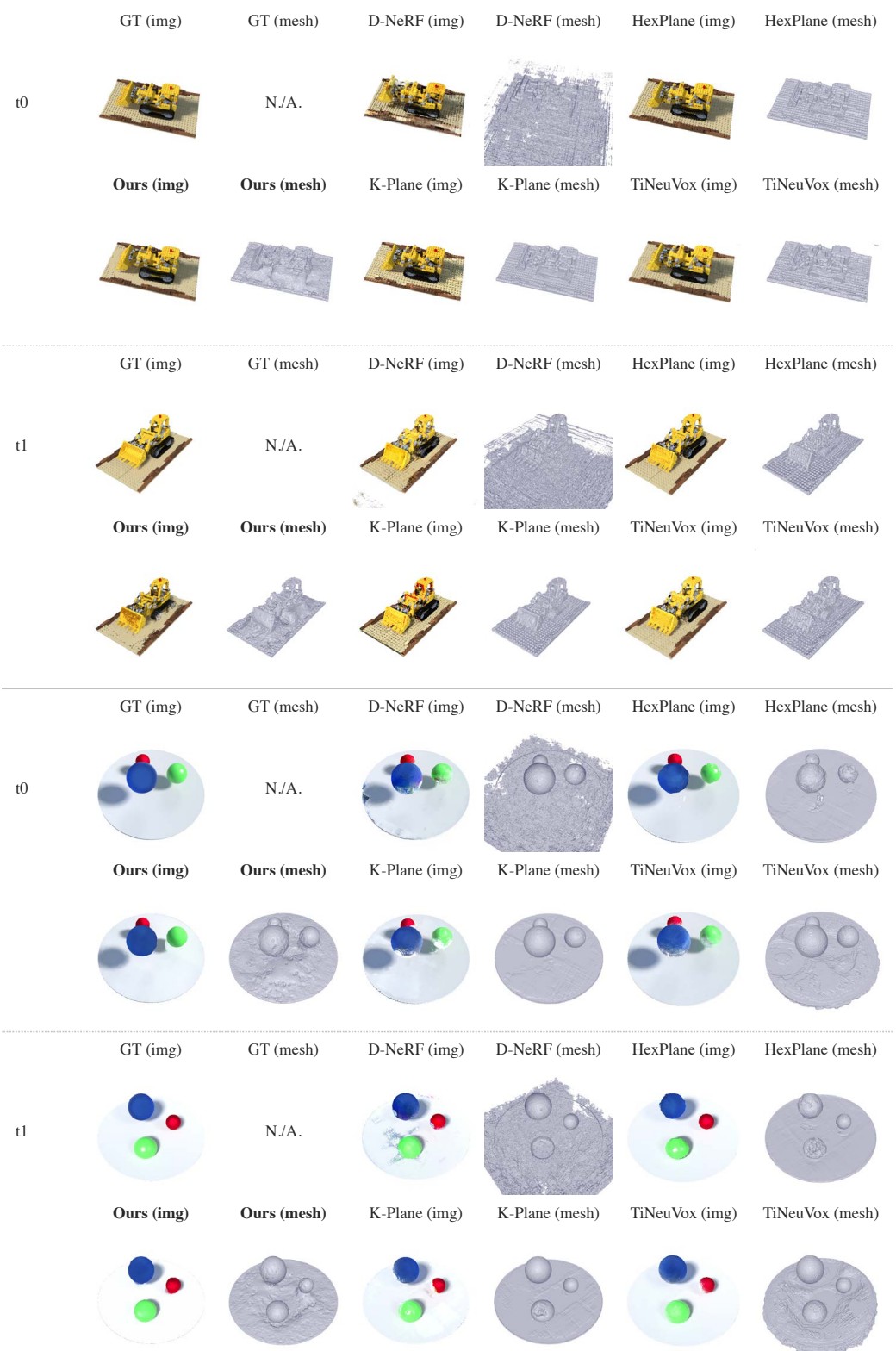

Figure 20: Results comparison on the D-NeRF dataset. We show the reconstructed mesh and the mesh rendering image. Our method reconstructs better geometry and appearance than other baselines.

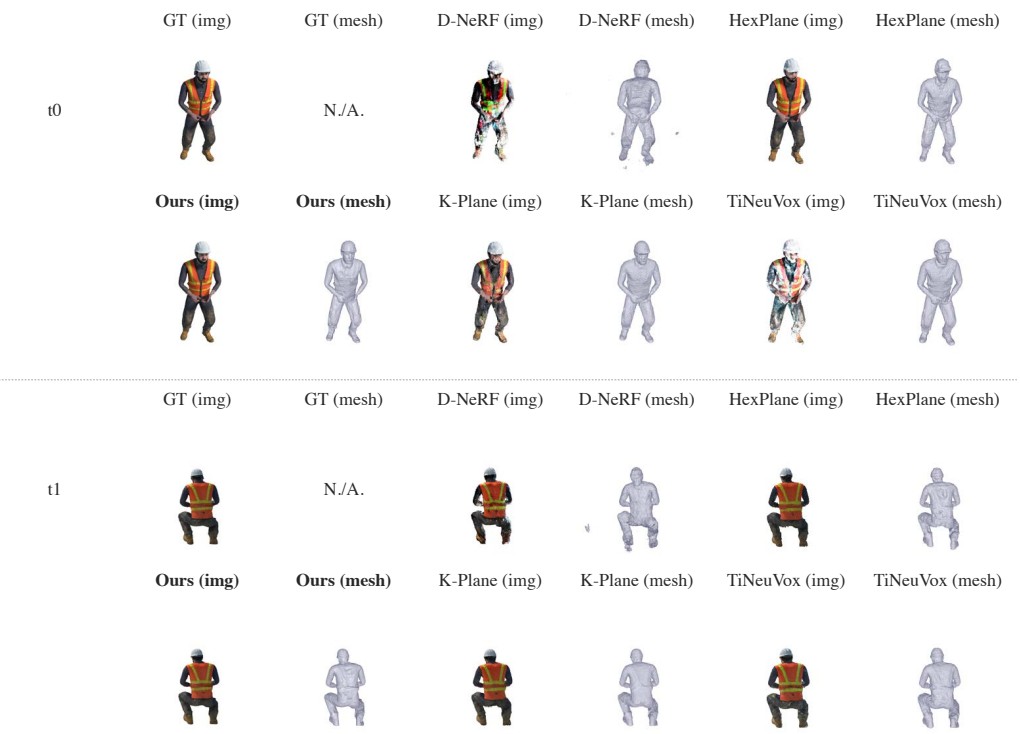

Figure 21: Results comparison on the D-NeRF dataset. We show the reconstructed mesh and the mesh rendering image. Our method reconstructs better geometry and appearance than other baselines.

## M   MORE QUALITATIVE RESULTS ON THE DG-MESH DATASET

We present the mesh reconstruction results of DG-Mesh across all six samples in the DG-Mesh dataset. Each sample consists of 200 frames of a moving object, with ground truth camera parameters, corresponding images, and ground truth mesh for each time frame. The camera views are uniformly distributed around the target objects, either on a full sphere or an upper hemisphere. For each sample, we provide visualizations of the reconstructed mesh along with mesh surface rendering results at two different time frames, $t0$ and $t1$. We compare these visual results against several baselines, including D-NeRF (Pumarola et al., 2021), HexPlane (Cao & Johnson, 2023), K-Plane (Fridovich-Keil et al., 2023), and TiNeuVox (Fang et al., 2022). The results demonstrate that our method achieves superior mesh reconstruction and rendering quality. Notably, DG-Mesh excels in reconstructing high-quality mesh surfaces, even in challenging regions such as a bird's wings or a horse's tail, where fine details and complex motions are involved.

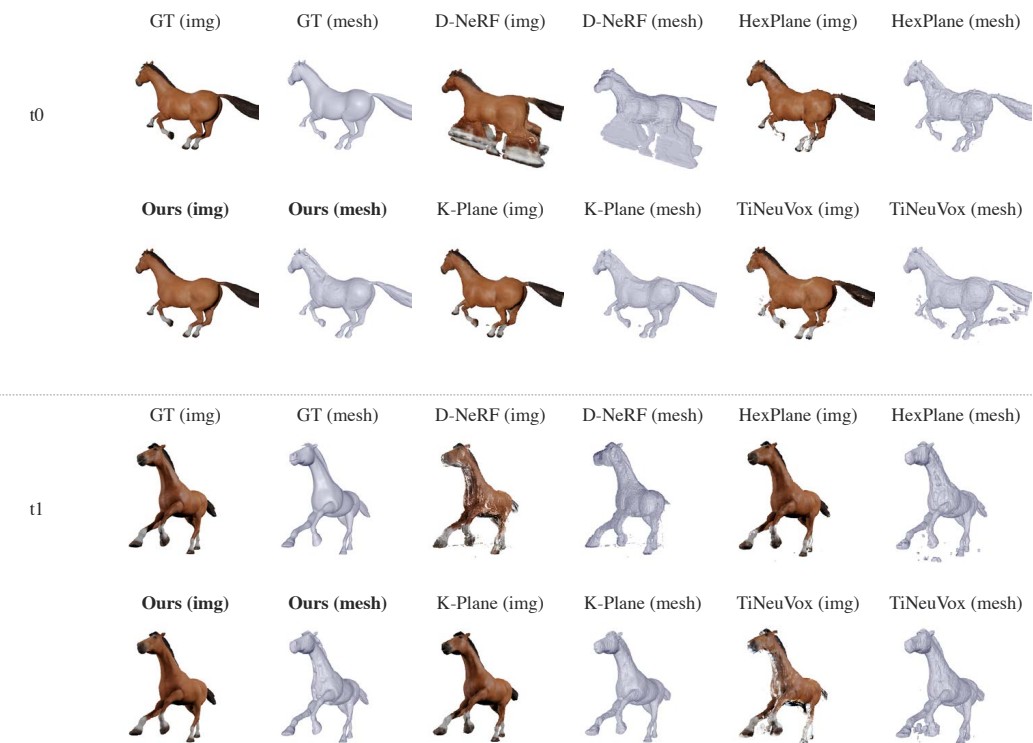

Figure 22: Results comparison on the DG-Mesh dataset. We show the reconstructed mesh and the mesh rendering image. Our method reconstructs better geometry and appearance than other baselines.

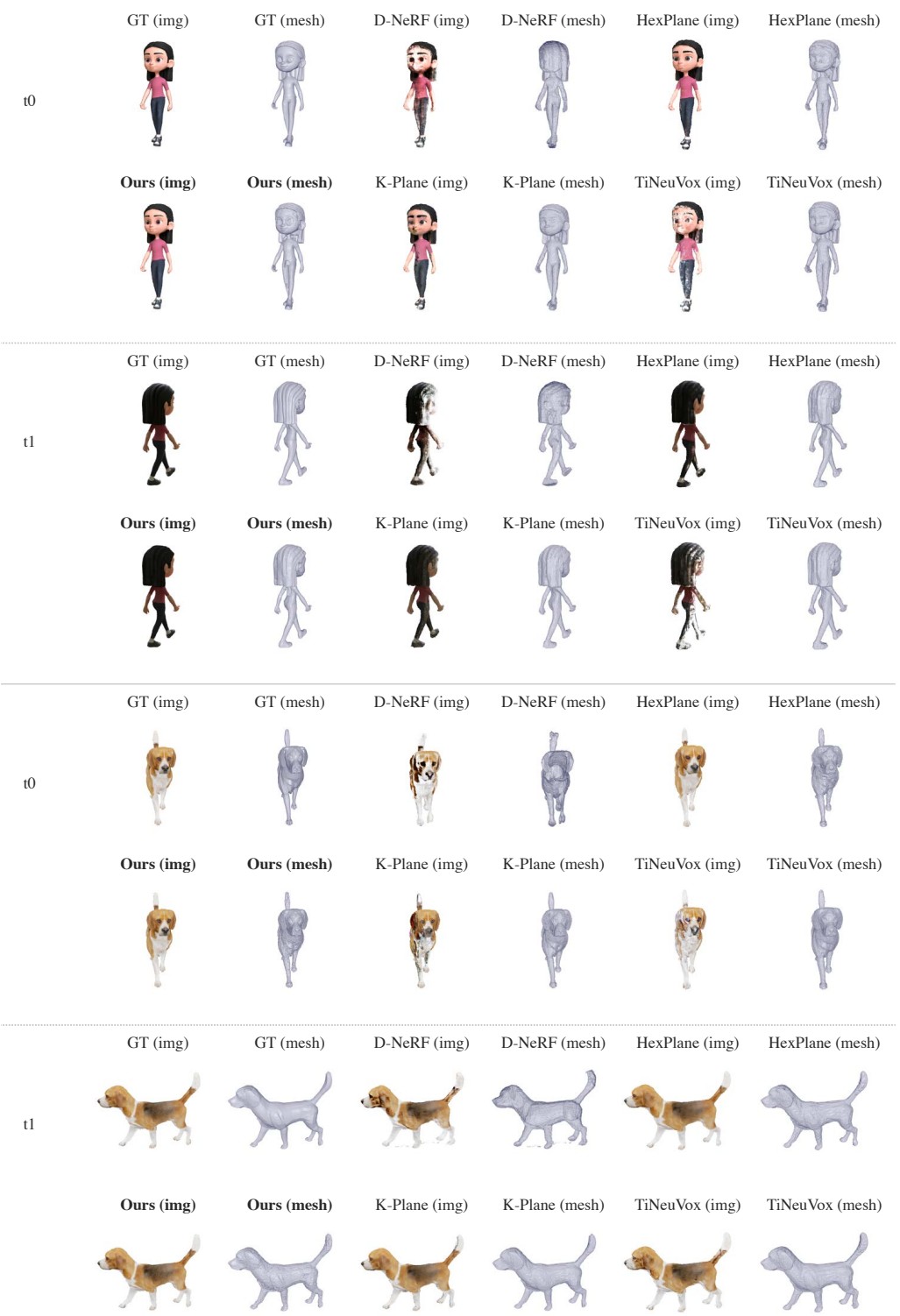

Figure 23: Results comparison on the DG-Mesh dataset. We show the reconstructed mesh and the mesh rendering image. Our method reconstructs better geometry and appearance than other baselines.

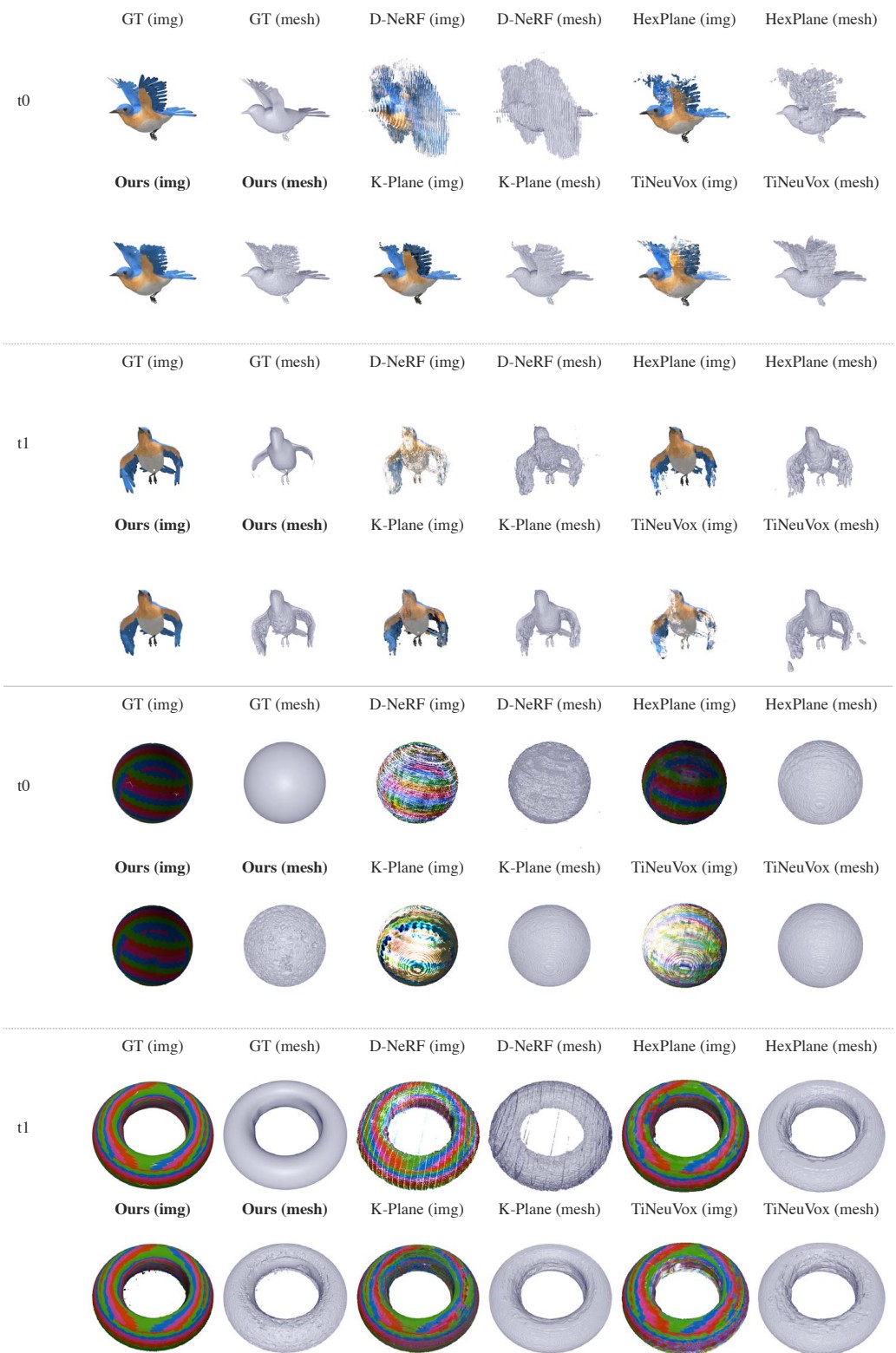

Figure 24: Results comparison on the DG-Mesh dataset. We show the reconstructed mesh and the mesh rendering image. Our method reconstructs better geometry and appearance than other baselines.

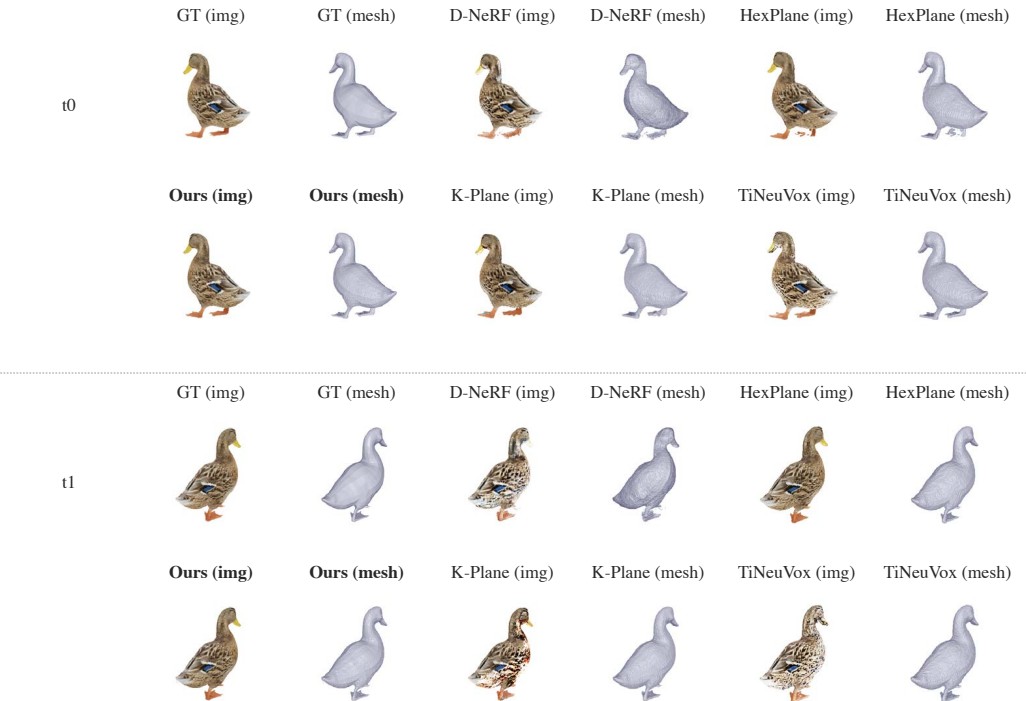

Figure 25: Results comparison on the DG-Mesh dataset. We show the reconstructed mesh and the mesh rendering image. Our method reconstructs better geometry and appearance than other baselines.

## N    APPLICATION EXAMPLES

Mesh remains the predominant representation supported by many physics simulators and rendering engines, making it a versatile tool for various applications. DG-Mesh provides dense correspondence, which proves especially useful for downstream tasks like shape manipulation and texture editing.

We demonstrate two examples of mesh editing enabled by our method, as illustrated in Figure 26. One application involves inserting the extracted mesh into a new scene, where ray-tracing can be performed. Another application is dynamic texture editing. Using the time-consistent mesh in the first frame, we can edit the vertex colors by painting them directly onto the surface. This modification is then propagated consistently across subsequent frames, maintaining the same color and pattern for the corresponding vertices. The entire editing process is intuitive and user-friendly, making it accessible even for complex tasks.

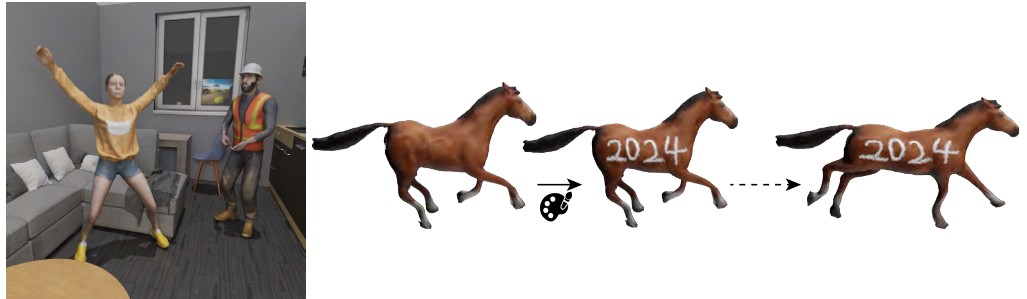

Figure 26: Two types of application can be done with time-consistent mesh: 1. Ray-tracing in the rendering engine. 2. Texture editing in dynamic object: With correspondence across time frames, we just need to edit the first frame in a sequence and the change will be automatically applied to the rest of the sequence.

