# OpenReview forum: "Dynamic Gaussians Mesh: Consistent Mesh Reconstruction from Dynamic Scenes"
_ICLR.cc/2025/Conference — ICLR 2025 Poster_

### Official Review · Reviewer_9ERR · 2024-10-26

**Soundness:** 3
**Presentation:** 4
**Contribution:** 3
**Rating:** 6
**Confidence:** 5

**Summary:**

This paper proposes a method to extract high-quality mesh from image inputs in monocular/ efficient multiview setups. The authors propose a Gaussian-Mesh anchoring method and cycle-consistent deformation loss to enhance the upper bounds of DG-Mesh representations. The authors propose many downstream tasks to demonstrate the widely applications of DG-Mesh.

**Strengths:**

1. This study presents a novel framework that enables the reconstruction of high-fidelity, time-consistent meshes from monocular videos. Generating meshes for dynamic scenes is a challenging yet critical task, and the author makes a significant contribution in addressing it.

2. The proposed approach includes a Gaussian-Mesh anchoring technique that produces uniformly distributed Gaussians in each frame, enhancing the effectiveness of mesh reconstruction. Cycle-consistent deformation loss adjusts the deformed Gaussians, offering a new method for densification in deformation-based 3D Gaussians.

**Weaknesses:**

1. **Baseline Selection:** I suggest that the authors consider incorporating additional baselines such as Spacetime-GS (Li et al., CVPR 2024), 4DGS (Yang et al., ICLR 2024), and 4D-GS (Wu et al., CVPR 2024), given the recent advancements in Dynamic Gaussian representations. It would also be helpful to include some NeRF baselines in the supplementary materials.

2. **Color Prediction:** As mentioned in Section 4.2, the mesh is extracted using Nvdiffrast. How do the authors handle the color extraction for the mesh? Is the DG-Mesh representation used for this purpose? If so, why? I find it challenging to apply this approach in real-world scenarios.

3. **Real-World Monocular Scenes:** In my opinion, DG-Mesh is still difficult to apply in real-world settings, as the original 4D representations struggle to maintain multi-view space-time consistency. While the results appear satisfactory in training views, they might deteriorate in novel views or the final extracted mesh. I recommend that the authors emphasize the robustness of multi-view videos rather than focusing solely on monocular videos. In terms of monocular dynamic scene reconstruction, there are several relevant works (e.g., MoSca: Dynamic Gaussian Fusion from Casual Videos via 4D Motion Scaffolds, Dynamic Gaussian Marbles, Shape of Motion, Dreamscene4D, MoDGS) that have yet to deliver satisfactory results. Claiming effectiveness in monocular videos might be an overstatement.

**Questions:**

Please refer to weakness.
Overall, I think this is a pioneer work and My main consideration is weakness2,3.

---

> ### Author Response · Authors · 2024-11-22
> **Response to Reviewer 9ERR (1/2)**
>
> We thank the reviewer for their valuable feedback. We address your comments in the following.
>
> ----
>
> **Q:** *Incorporating additional baselines such as Spacetime-GS, 4DGS, and 4D-GS. Including some NeRF baselines in the supplementary materials.*
>
> **A:** Please refer to our *General Response (1)*, where we have added baseline comparisons with 4DGS [1]. The Spacetime-GS [2] fails on the D-NeRF and DG-Mesh datasets because it assumes multi-view inputs, which are not available in these datasets. For 4D-GS [3] and other NeRF baselines (e.g., D-NeRF, HexPlane, K-Plane, and TiNeuVox), **we have already included comparisons in our submission**. Their quantitative results are presented in Tables 1 and 2 of the paper, with qualitative results provided in the supplementary material Section I and J. DG-Mesh consistently reconstructs better mesh quality compared to these methods.
>
> ---
>
> **Q:** *The mesh is extracted using Nvdiffrast. How to handle the color extraction for the mesh? Is the DG-Mesh representation used for this purpose? If so, why?*
>
> **A:** We do not use Nvdiffrast to extract meshes. The mesh surface geometry is extracted through DPSR and Diff.M.C. After which Nvdiffrast is applied to only rasterize the mesh vertices' color into rendered images. The vertex color is derived from a learned appearance MLP, which takes the vertex's position in canonical space and the time label as inputs, producing the corresponding vertex color. For more details about the color implementation, please refer to our *General Response (2)*. The DG-Mesh surface rendering enables the network to learn appearance directly and precisely at the object's surface, addressing limitations of volume rendering-based methods (e.g., 4D-GS, HexPlane). These methods often prioritize visual appearance, potentially storing appearance information at positions away from the actual surface, which can lead to inaccuracies in surface-aware tasks.
>
> ---
>
> **Q:** *DG-Mesh is still difficult to apply in monocular real-world settings, as the original 4D representations struggle to maintain multi-view space-time consistency. While the results appear satisfactory in training views, they might deteriorate in novel views or the final extracted mesh.*
>
> **A:** We have presented **significant novel-view rendering results** under monocular real-world settings in our submission. Specifically, we shown a comprehensive evaluation of the Nerfies [4] dataset, which includes real monocular videos captured by smartphones. The mesh reconstruction and rendering results for testing views are presented in Figure 7 and Table 5 of the paper. We have also shown real video performance in these [[videos]](https://dgmesh.github.io/#real_res). These results demonstrate that DG-Mesh **performs effectively** under real-world scenarios, maintaining high-quality mesh reconstruction and rendering consistency in novel views.
>
> DG-Mesh addresses the limitations of the original 4D representation in maintaining multi-view space-time consistency through the following key modules:
> * **Surface-Guided Gaussian Optimization**: The integration of mesh optimization and Gaussian-Mesh Anchoring acts as **a surface constraint on the original 4D Gaussian representation**. This ensures that the Gaussian distribution aligns with the surface topology, improving the multi-view consistency.
> * **Cycle-Consistency Deformation**: Enhances temporal coherence across all frames, ensuring that the Gaussians remain consistent in both space and time.

---

> ### Author Response · Authors · 2024-11-22
> **Response to Reviewer 9ERR (2/2)**
>
> **Q:** *I recommend that the authors emphasize the robustness of multi-view videos rather than focusing solely on monocular videos.*
>
> **A:** Thank you for your suggestion. We would like to clarify that our paper **does not solely focus on monocular videos**. In fact, we have demonstrated promising results under multi-view video setups, as presented in Section G of the supplementary material. However, multi-view setups are often hard to acquire in real-world scenarios, as they require complex capture systems. In contrast, our approach is designed to work effectively with everyday videos captured using a single hand-held camera, making it more applicable to real-world use cases.
>
> ---
>
> **Q:** *In terms of monocular dynamic scene reconstruction, there are several relevant works (e.g., MoSca: Dynamic Gaussian Fusion from Casual Videos via 4D Motion Scaffolds, Dynamic Gaussian Marbles, Shape of Motion, Dreamscene4D, MoDGS) that have yet to deliver satisfactory results. Claiming effectiveness in monocular videos might be an overstatement.*
>
> **A:** We appreciate the reviewer’s comment and wholeheartedly agree that monocular dynamic scene reconstruction remains a highly challenging problem. While existing works have made notable progress, they still face significant limitations in achieving space-time consistent reconstruction. However, unlike these methods, our work focuses on a different task. Rather than prioritizing novel view synthesis, we center our efforts on reconstructing the foreground object's mesh, **which is a more achievable goal**. By encouraging a time-consistent mesh, our approach ensures improved Gaussian representation and geometric consistency.
>
> We respectfully disagree with the reviewer’s assessment that our claim of effectiveness in monocular videos is an overstatement. **We have demonstrated compelling results as evidence, successfully constructing time-consistent meshes using everyday videos captured with simple hand-held devices. To the best of our knowledge, no prior work has accomplished this**. We do agree that our work does not fully solve the complex problem of dynamic mesh reconstruction from monocular videos. However, we strongly believe that our methods, evaluation protocol, and dataset represent a significant step forward. Our contributions establish a critical foundation and offer valuable tools for the community.
>
> ---
>
> Please do not hesitate to let us know if you have any additional comments.
>
> **Reference**
>
> [1] *Yang, Zeyu, et al. "Real-time photorealistic dynamic scene representation and rendering with 4d gaussian splatting." ICLR, 2024.*
>
> [2] *Huang, Yi-Hua, et al. "Sc-gs: Sparse-controlled gaussian splatting for editable dynamic scenes." CVPR, 2024.*
>
> [3] *Wu, Guanjun, et al. "4d gaussian splatting for real-time dynamic scene rendering." CVPR, 2024.*
>
> [4] *Park, Keunhong, et al. "Nerfies: Deformable neural radiance fields." ICCV, 2021.*
>
> [5] *Liu, Lingjie, et al. "Neural actor: Neural free-view synthesis of human actors with pose control." TOG, 2021.*

---

> > ### Comment · Reviewer_9ERR · 2024-11-22
> > **Thanks for your feedback**
> >
> > Dear authors,
> > Thanks for your comprehensive feedback, I still have several problems:
> >
> > 1. monocular dynamic scenes: If DG-Mesh wants to claim that this method could be applied to the monocular scenes, please provide foreground mesh's results on the monocular dataset with the spatial-temporal consistency (e.g. following dycheck's setups, or DAVIS dataset). Show the results on the views the same as the input video is just an interpolation. I doubt that all the deformable GS methods can only overfit on the training views, so how can DG-Mesh get a precise mesh under that case? I think claiming that **monocular** is a strict statement and I kindly request authors consider that. And what's more, could the author explain what you contribute to the monocular dynamic scenes? What are the core problems of extracting meshes from the *monocular* input and how do you solve them? I just think your proposed method is more like a general method instead of being designed for monocular videos only. Since the author also acknowledge that DG-Mesh can solve multiview inputs, why don't change the paper's name to *DYNAMIC GAUSSIANSMESH : CONSISTENT MESH RECONSTRUCTION FROM **DYNAMIC SCENES**?*
> >
> >
> > 2. Color prediction. Mesh is a final representation that many files would accept such as Blender. If a user use the MLP+mesh, how could he/she import that?

---

> > > ### Author Response · Authors · 2024-11-24
> > > **Response to Reviewer 9ERR**
> > >
> > > Dear Reviewer 9ERR,
> > >
> > > Thank you for your feedback. Regarding the monocular setup, we follow Dycheck’s setup and provide the DG-Mesh results on their dataset here ([#video1](https://dgmesh.github.io/rebuttal_materials/dycheck/paper-windmill.mp4), [#video2](https://dgmesh.github.io/rebuttal_materials/dycheck/haru-sit.mp4), [#video3](https://dgmesh.github.io/rebuttal_materials/dycheck/mochi-high-five.mp4)). We acknowledge that similar to previous dynamic Gaussians methods, we encounter limitations in cases where object motion is too extreme or segmentation is imperfect ([#video4](https://dgmesh.github.io/rebuttal_materials/dycheck/sriracha-tree.mp4), [#video5](https://dgmesh.github.io/rebuttal_materials/dycheck/backpack.mp4)). We also want to clarify that our previously shown real-world results were captured under testing views, some of which have significant baseline differences from the input views.
> > >
> > > While our approach provides promising dynamic geometry reconstruction results under certain monocular setups, we concur with your opinion that our method is not explicitly tailored for monocular videos, and we will revise the paper's title as you suggested to better reflect the scope and focus of the method.
> > >
> > > As for color prediction, when exporting mesh, our method queries the MLP and stores the color information directly in the mesh vertices. This colored mesh can then be imported into standard 3D software such as Blender or other rendering engines.
> > >
> > > We hope that our clarifications have addressed your concerns and convinced you to consider raising your score. Thank you again for your thoughtful comments and suggestions!

---

> ### Comment · Reviewer_9ERR · 2024-11-25
> **I will keep my score**
>
> Dear authors,
>
> I acknowledge that DG-Mesh is the one of the first work to extract high quality meshes with 3D Gaussians, the novelty of this paper is enough.
>
> However, your basic representation, deformable 3D-GS, is not designed for monocular input, and both papers seem didn't claim the special contribution for the monocular videos (results are also not convincible).
>
> I hope authors could consider following ideas:
>
> 1. Replacing D-3DGS with more state-of-the-art representations, such as MoSca(Lei et al), shape-of-motion(Wang et al), add more supervision (such as flow, tracking) to fit **monocular** setups.
>
> 2. Claiming DG-Mesh's **wider utility**, providing the results on multiview's dataset: Neu3D and other datasets on the spacetime-GS.
>
> 3. Demonstrate that proposed cycle consitency deformation and other method can be used in that representations, DG-Mesh could be an efficient plugin.

---

> > ### Author Response · Authors · 2024-11-25
> > **Response to Reviewer 9ERR**
> >
> > Dear Reviewer 9ERR,
> >
> > Thank you for acknowledging the novelty of this paper. However, we respectfully ask the reviewer to judge our results based on scientific evidence instead of personal feelings, or just “not convincible”. Otherwise, we believe the claim is unfair.
> >
> > The reviewer simply has no evidence showing that there exists better results than those provided by our method. On the other hand, in our attached video we show **our method achieves the best mesh reconstruction results compared to Deformable-GS**. Please provide the paper that has tackled the same task on the same dataset with better mesh reconstruction results. Otherwise, dismissing the results as unconvincing seems unwarranted.
> >
> > When RCNN was first proposed in object detection, it only performed 40% in the Pascal dataset instead of the 80% we are seeing now. But that is an important result that lays the foundation for object detection today. Here, it is similar; while “the first work” (as acknowledged by the reviewer) in this field will not just directly solve the problem, that is normal, but it will provide the foundation for future research.
> >
> > **Regarding the scope of the work**: Our previous answer already revised the paper title and claims to better reflect the scope of our contributions, not just for monocular video. The submission now includes thorough experiments on both monocular and multi-view datasets.

---

> ### Comment · Reviewer_9ERR · 2024-11-25
> **Change my score to weak accept**
>
> Dear authors,
>
> Thanks for your explaination. After thorough consideration and read your rebuttal, I decide to change my score to WA and wait authors' revision on the full paper.
>
> Since there are many following works (Dynamic 2D Gaussians (zhang et al), spacetime 2D Gaussians (wang et al)), DG-Mesh may be an solid baseline for dynamic mesh reconstruction.

---

> > ### Author Response · Authors · 2024-11-26
> > **Response to Reviewer 9ERR**
> >
> > Dear Reviewer 9ERR,
> >
> > Thank you for reconsidering our work and raising your score. We have uploaded the revised paper for your reference.

---

### Official Review · Reviewer_mKQh · 2024-11-02

**Soundness:** 2
**Presentation:** 3
**Contribution:** 2
**Rating:** 8
**Confidence:** 5

**Summary:**

The paper introduces a method for using deformable Gaussian Splatting to generate dynamic meshes for each frame of a monocular video. It employs a set of canonical Gaussians, deforming them at each timestep with an MLP. These deformed Gaussians are then used to construct a mesh for each frame through Poisson Reconstruction and Marching Cubes. The mesh serves as an anchor for the Gaussians, ensuring a one-to-one correspondence between mesh faces and Gaussians. Instead of textures, vertex colors are applied to the mesh, which is rendered as an image for supervision.

The paper is clear and straightforward, though it does have some limitations. The resulting mesh is a per-frame set rather than a truly deformable mesh. Additionally, the mesh is colored using vertex colors instead of a high-resolution UV map. These constraints raise concerns about the method’s applicability.

**Strengths:**

The paper sets an ambitious goal: reconstructing dynamic meshes from monocular videos, a critical challenge in graphics and vision. It begins with a baseline approach using Deformable GS to model the dynamic scene. Mesh reconstruction is then applied to each frame, incorporating an anchoring constraint on the Gaussians through the mesh. This approach helps achieve a smooth surface but can adversely affect the rendering results.

**Weaknesses:**

Here are the weaknesses of the paper:

1. **Mesh Representation:** The method produces a set of per-frame meshes rather than a deformable mesh. Although mesh representations are known for their rich operations and memory efficiency, the paper does not demonstrate how to register each frame to a unified mesh. This per-frame approach is not memory-efficient and makes editing geometry and textures cumbersome, as changes cannot be easily propagated across frames without manual effort.

2. **Detail Preservation:** The mesh optimization and anchoring mechanism do not preserve details effectively, as demonstrated in Table 1. While meshes efficiently represent low-frequency geometry, high-frequency textures should ideally be stored in a UV map. This method colors the mesh at vertices, limiting its ability to capture high-frequency details. The one-to-one mapping between mesh faces and Gaussians necessitates a high-resolution mesh to accommodate the large number of Gaussians needed for rich texture and geometry representation. However, the mesh number is limited by the resolution of Marching Cubes and is hard to increase as much as needed Gaussians. This results in inefficient memory use, as mesh faces are not effectively utilized for storing detailed textures.

3. **Missing Related Works:** The paper omits several closely related works:
   - "SC-GS: Sparse-Controlled Gaussian Splatting for Editable Dynamic Scenes" by Huang et al.
   - "Spacetime Gaussian Feature Splatting for Real-Time Dynamic View Synthesis" by Li et al.
   - "Neural Parametric Gaussians for Monocular Non-Rigid Object Reconstruction" by Das et al.

   It is recommended to compare with **SC-GS** in Table 1, as it is the current state-of-the-art method in this area. It's not necessary to overwhelm its result since the focus of the two papers differs.

**Questions:**

1. Is there a way to unify the per-frame meshes into a deformable mesh? Once unified, this mesh could be optimized with a UV map to capture rich texture details. Neural temporal textures might also be useful, as they can address inaccuracies in deformation correspondence across frames by providing time-dependent texture adjustments.

2. Can DG-Mesh be used to obtain a skeleton and skinned mesh? Extracting animatable assets from dynamic videos is crucial. This could be suggested as a potential enhancement in the future work section. The paper can articulate how it advances the field toward this goal to some extent.

---

> ### Author Response · Authors · 2024-11-22
> **Response to Reviewer mKQh**
>
> We thank the reviewer for their valuable feedback. We address your comments in the following.
>
> ---
>
> **Q:** *The paper does not demonstrate how to register each frame to a unified mesh. This per-frame approach is not memory-efficient and makes editing geometry and textures cumbersome, as changes cannot be easily propagated across frames without manual effort.*
>
> **A:** We do **NOT** store pre-frame meshes, please refer to Section 3.3 of our submission. Instead, we maintain a canonical Gaussian set and a set of compact MLPs and can achieve relatively fast real-time mesh querying through Diff.M.C. and DPSR. Geometry and texture editing can be performed on either the canonical or deformed Gaussians, **with changes easily propagated across frames via the correspondence established by the Cycle-Consistent deformation networks**.
>
> To address the reviewer's concern about memory efficiency, we compare the memory usage of DG-Mesh with other data structure storage methods below. DG-Mesh demonstrates significantly reduced storage requirements compared to per-frame meshes, with memory size comparable to unified mesh approaches.
>
> | | Unified Mesh | Per-frame Mesh (200 frames) | DG-Mesh (Ours) |
> |:---:|:---:|:---:|:---:|
> | Mem. Size | 5.3 MB | 600 MB | 28.1 MB |
>
> ---
>
> **Q:** *The mesh optimization and anchoring mechanism do not preserve details effectively, as demonstrated in Table 1. This method colors the mesh at vertices, limiting its ability to capture high-frequency details.*
>
> **A:** We achieve **the best mesh rendering and surface reconstruction results** in Table 1. Moreover, Figures 5 and 6 of the submission clearly demonstrate that we preserve **better geometry details** than other baselines. The ability of vertex color to capture high-frequency details is constrained primarily only when the vertex density is low, leading to noticeable color interpolation artifacts across faces. Our method employs a high grid resolution during iso-surface extraction, which ensures the extracted mesh has a sufficient vertex density, allowing it to effectively preserve and store fine appearance details.
>
> ---
>
> **Q:** *The one-to-one mapping between mesh faces and Gaussians necessitates a high-resolution mesh to accommodate the large number of Gaussians needed for rich texture and geometry representation. However, the mesh number is limited by the resolution of Marching Cubes and is hard to increase as much as needed Gaussians.*
>
> **A:** Our method does not require an excessive number of Gaussians to store rich texture details, as **our primary goal is mesh reconstruction, not the rendering quality of Gaussians**. The current grid resolution of 288 is sufficient to produce enough mesh faces to effectively restore surface appearance in general reconstruction tasks. Additionally, this grid resolution can be easily scaled up to handle more complex scenes if required.
>
> ---
>
> **Q:** *Missing related works and compare with SC-GS.*
>
> **A:** Please refer to our *General Response (1)*, where we provide the baseline comparison with SC-GS, Spacetime-GS, and other recent dynamic Gaussian methods. DG-Mesh consistently demonstrates superior mesh reconstruction quality compared to these baseline methods. We will include these additional comparisons in the updated version of our submission.
>
> ---
>
> **Q:** *Consider unifying the per-frame meshes into a deformable mesh and optimize a neural temporal UV map.*
>
> **A:**  **Deformable meshes (i.e., template-based meshes) CANNOT accommodate topology changes during deformation**. As illustrated in the [[Image]](https://dgmesh.github.io/rebuttal_materials/tempalte_fail.png), a common failure case occurs when attempting to deform a spherical template mesh into a torus, indicating the inflexibility of unified meshes in adapting face connectivity. In contrast, our point-based representation circumvents the need to manage face connectivity during deformation, offering greater adaptability. We discuss the limitations of using a template mesh in detail in Section C of our supplementary material.
>
> ---
>
> **Q:** *Can DG-Mesh be used to obtain a skeleton and skinned mesh?*
>
> **A:** DG-Mesh can absolutely be used to obtain a skeleton and skinned mesh. The reconstructed mesh sequence from video input can be integrated with auto-rigging methods to animate dynamic content. Using DG-Mesh, we reconstruct the dynamic content [[video result]](https://dgmesh.github.io/rebuttal_materials/animation/dgmesh.mp4) and re-animate it by extracting skeleton and surface skinning properties from our DG-Mesh representation [[animation video]](https://dgmesh.github.io/rebuttal_materials/animation/animation.mp4). This demonstrates the broader applicability and versatility of our method in animation and related tasks.
>
> ---
>
> Please do not hesitate to let us know if you have any additional comments.

---

> > ### Comment · Reviewer_mKQh · 2024-11-22
> >
> > Thank you to the authors for their detailed response. It addressed many of my concerns regarding the work. I greatly appreciate the animation results in the accompanying video, which played a significant role in my decision to raise the score.
> >
> > Regarding the discussion on per-frame mesh representation, I still believe that a unified deformable mesh template is superior to the current regularized dynamic Gaussians. Topology changes are relatively uncommon and can be excluded from the research scope when studying dynamic geometry. The DGMesh is essentially an intermediate representation, with the final output being per-frame varying template meshes. Therefore, I maintain that if a deformable template and a compact deformation basis could be learned, the paper would achieve a higher level of excellence.
> >
> > That said, I can see the progress made toward this goal in the attached rebuttal video. If the skinning step were further optimized and a template representation was incorporated into DGMesh, the approach would be even stronger. Nonetheless, to some extent, this paper advances the field toward that goal, and I agree with 9err that it is pioneering in this domain.
> >
> > My decision is to raise the score to 6. The exploration and efforts in dynamic geometry reconstruction should be encouraged, and we must remain patient with incremental progress in this challenging area.

---

> > > ### Author Response · Authors · 2024-11-24
> > > **Response to Reviewer mKQh**
> > >
> > > Dear Reviewer mKQh,
> > >
> > > Thank you for thoroughly reviewing our rebuttal materials and animation results with such care and attention! We are grateful for your recognition of our efforts and the pioneering aspects of our work in dynamic geometry reconstruction. We are deeply inspired by your feedback to extract the skeleton and skinning from DG-Mesh and see the broader impact of combining skeleton/template representation with the DG-Mesh pipeline. This direction will definitely be explored in our future work.

---

### Official Review · Reviewer_wAGq · 2024-11-03

**Soundness:** 3
**Presentation:** 3
**Contribution:** 3
**Rating:** 6
**Confidence:** 4

**Summary:**

The paper introduces Dynamic Gaussians Mesh (DG-Mesh), a novel framework designed to reconstruct time-consistent, high-fidelity meshes from monocular video footage. Utilizing 3D Gaussian Splatting (3DGS), the framework ensures temporal consistency and superior mesh reconstruction quality. Innovations such as Gaussian-Mesh Anchoring and cycle-consistent deformation significantly enhance the distribution and stability of 3D Gaussians across frames.

**Strengths:**

1. The manuscript is well-structured, with fluent writing and a clear explanation of methods, making it easy for readers to understand.

2. This method combines the advantages of Mesh and 3D GS and achieves remarkable improvements in surface detail and rendering quality in dynamic object reconstruction.

3. The experiment is thorough, with comprehensive benchmarks against established methods, demonstrating significant enhancements in both mesh quality and image rendering.

**Weaknesses:**

1. The integration of 3D Gaussians with mesh surfaces is good, yet the paper does not explore the potential benefits of using 2D Gaussian Splatting for surface and image reconstruction accuracy. Could a 2D representation be more effective or efficient when Gaussians lie on the mesh surface?

2. The method involves separate rasterization processes for Mesh and 3D Gaussians. However, the paper lacks a detailed explanation of how color information is harmonized between these components. How is color from the 3DGS sh presentation passed to the mesh during rasterization?

3. Binding a single Gaussian to each mesh face raises questions about the distribution logic. Why was this strategy chosen? Did the authors consider alternatives, such as placing multiple Gaussians on a single mesh face or varying the number of Gaussians based on mesh face properties?

4. Table 1 only presents PSNRm metrics; however, standard PSNR metrics for direct 3DGS rendering results are absent. Could the authors clarify the reason for this omission?

5. The paper does not address how varying the number of mesh surfaces extracted from the same scene affects rendering accuracy and computational efficiency.

**Questions:**

Please see the weaknesses part.

---

> ### Author Response · Authors · 2024-11-22
> **Response to Reviewer wAGq**
>
> We thank the reviewer for their valuable feedback. We address your comments in the following.
>
> ---
>
> **Q:** *Could 2DGS be more effective/efficient when Gaussians lie on the mesh surface?*
>
> **A:** We replace the 3DGS in our pipeline with 2DGS by enforcing one dimension of the original 3DGS to be zero. Compared to 3DGS, 2DGS generates more redundant 2D Gaussian plates to represent the geometry and appearance, which significantly increases the computational load. The use of 2DGS did not result in noticeable improvements in performance, suggesting that 3DGS remains a more effective and efficient choice for our pipeline.
>
> | | PSNR ↑ | PSNR_m ↑ | CD (10⁻²) ↓ | EMD(10⁻¹) ↓ | GS # | Inference Speed ↑ |
> |:---:|:---:|:---:|:---:|:---:|:---:|:---:|
> | 2DGS | 24.044 | 30.531 | 0.316 | 1.423 | 62394 | 7.835 FPS |
> | 3DGS | 25.245 | 30.668 | 0.306 | 1.298 | 51773 | 10.539 FPS |
>
> ---
>
> **Q:** *How is color from the 3DGS sh presentation passed to the mesh during rasterization?*
>
> **A:** Please refer to our *General Response (2)*. DG-Mesh manages appearance separately for Gaussian rasterization and mesh rasterization. Gaussian rasterization derives colors from the volume rendering results of spherical harmonics (SH) stored on each Gaussian, while mesh rasterization uses a learned appearance MLP to predict time-dependent surface colors.
>
> ---
>
> **Q:** *Why choose to bind a single Gaussian to each mesh face? Why not place multiple GS on a single mesh face or vary the number of GS based on mesh face properties?*
>
> **A:** Compared to placing multiple GS on a single mesh face, binding a single GS to the **centroid** of each mesh face ensures **a more uniform Gaussian distribution**. Placing multiple GS inside a triangular face requires a specifically tailored approach for different quantities to ensure uniformity across the face, which reduces the generalizability of the method. Additionally, our results show that adding more GS does not guarantee improved reconstruction quality. We evaluated the performance and querying speed by varying the number of GS bound to each mesh face, and the quantitative results demonstrate that placing multiple GS did not enhance reconstruction quality while resulting in slower anchoring speed. This highlights the efficiency and effectiveness of binding a single GS per face.
>
> | GS Binded | CD (10⁻²) ↓ | EMD (10⁻¹) ↓ | GS # | Speed |
> |:---:|:---:|:---:|:---:|:---:|
> | 1 | 0.306 | 1.298 | 51773 | 0.080s |
> | 3 | 0.313 | 1.300 | 54682 | 0.127s |
>
> ---
>
> **Q:** *Standard PSNR metrics for direct 3DGS rendering results are absent.*
>
> **A:** The main focus of this paper is **mesh reconstruction**, which is why we followed prior work on neural mesh reconstruction [1, 2] by concentrating on comparing mesh rendering quality in the original submission. In response to feedback, we have included the direct 3DGS rendering results in Table 1 in the updated version of the submission and provide the corresponding quantitative results here for further evaluation. DG-Mesh applies Gaussian-Mesh anchoring to ensure geometric correctness by eliminating Gaussians that do not lie on the surface. In contrast, other baseline methods (e.g., D-NeRF, K-Plane, HexPlane, TiNeuVox-B) may store radiance at positions away from the surface or densify Gaussians at non-surface positions (e.g., 4DGS, Deformable-GS) to enhance visual quality. Consequently, DG-Mesh achieves superior surface reconstruction, with a slight trade-off in Gaussian rendering visual quality, and this trade-off is consistent with our objective of reconstructing the mesh and its appearance rather than focusing on the Gaussian's appearance.
>
> | Metrics | D-NeRF | K-Plane | HexPlane | TiNeuVox-B | 4DGS | Deformable-GS | DG-Mesh (Ours) |
> |:---:|:---:|:---:|:---:|:---:|:---:|:---:|:---:|
> | PSNR ↑ | 27.739 | 31.142 | 30.099 | 31.428 | 32.483 | 32.421 | 25.245 |
>
> ---
> **Q:** *Does not address how varying the number of mesh surfaces extracted from the same scene affects rendering accuracy and computational efficiency.*
>
> **A:** We conduct an ablation study on the grid resolution used during iso-surface extraction, which directly affects the number of mesh faces. The quantitative results below demonstrate that with a grid size of 288, our method achieves the best balance of rendering and geometric quality while maintaining reasonable inference speed.
>
> | Grid Res. | PSNR_m ↑ | CD (10⁻²) ↓ | EMD (10⁻¹) ↓ | Inference Speed (FPS) ↑ |
> |:---:|:---:|:---:|:---:|:---:|
> | 128 | 29.251 | 0.324 | 1.319 | **44.982** |
> | 192 | 30.210 | 0.322 | 1.305 | 25.482 |
> | 256 | 30.661 | 0.313 | 1.300 | 13.678 |
> | 288 | **30.668** | **0.306** | **1.298** | 10.539 |
>
> ---
>
> Please do not hesitate to let us know if you have any additional comments.
>
> **References**
>
> [1] *Wei, Xinyue, et al. "Neumanifold: Neural watertight manifold reconstruction with efficient and high-quality rendering support." WACV, 2025.*
>
> [2] *Yariv, Lior, et al. "Bakedsdf: Meshing neural sdfs for real-time view synthesis." ACM SIGGRAPH, 2023.*

---

> ### Author Response · Authors · 2024-11-24
> **Response to Reviewer wAGq**
>
> Dear Reviewer wAGq,
>
> Thank you again for your detailed feedback. As we approach the end of the author-reviewer discussion period, we notice that there have not yet been any responses to our rebuttal.
>
> Please feel free to request any additional information or clarification that may be needed. We hope to deliver all the information in time before the deadline.
>
> Thank you!

---

> > ### Comment · Reviewer_wAGq · 2024-11-25
> >
> > I appreciate the author's detailed response, which addressed some of my concerns. I raised my rating to 6.

---

> > > ### Author Response · Authors · 2024-11-26
> > > **Response to Reviewer wAGq**
> > >
> > > Dear Reviewer wAGq,
> > >
> > > Thank you for taking the time to review our responses. And we are glad that the questions have been resolved! These experiments and discussions have been included in our revised paper and supplement.

---

### Official Review · Reviewer_44mR · 2024-11-03

**Soundness:** 4
**Presentation:** 3
**Contribution:** 3
**Rating:** 8
**Confidence:** 4

**Summary:**

The proposed method uses two representations jointly to model a 3D object over time from forward-facing monocular video to produce state of the art results on publicly available datasets. The method produces a volumetric representation anchored to a mesh and can be used to track the deformation of the object over time. The tracked mesh can then be manipulated such as editing the texture. To achieve this, the proposed method uses Gaussians anchored to a canonical mesh using a forward / backward deformation field and Gaussian merging / creation rules based on mesh vertices. The method uses DPSR, DiffMC and laplacian regularization to generate intermediate meshes.

**Strengths:**

The paper is well written and follows a fairly intuitive and novel approach to generating both a deformable mesh and gaussian representation in parallel leveraging the benefits of both representations in the loss functions. The method performs at interactive speeds during inference while producing high quality results both qualitatively and quantitatively for both the mesh and Gaussian representations. The paper is evaluated on a variety of both real-world and synthetic datasets and produces tracked results using the forward / backward deformation of a canonical mesh. This is likely a significant improvement in learning deformable 3D scenes.

**Weaknesses:**

- The paper demonstrates that using both Gaussians and meshes improves the overall quality of novel view synthesis. There are several qualitative examples of higher quality meshes but it is unclear how / if appearance improves between the baselines and the proposed method when the underlying geometry is of similar quality.
- There is a lack of rigorous quantitative evaluation for several key decisions such as choosing the initialization, DMTet/DiffMC/FlexiCubes, mesh anchoring frequency and the training methodology and relies on qualitative observations. While these qualitative observations likely hold up well, having quantitative evaluations of various modules is desirable.

**Questions:**

- What's the intuition behind the improvement when using both Gaussian and mesh rendering losses?
- How would the method handle a synthetic large textured but planar surface such as a checkerboard where there might not be enough vertices to anchor gaussians to?
- In the attached supplementary materials, the anchored gaussians are represented as blue dots. What do these blue dots represent - is it just the mean positions?
- How is the "canonical" frame chosen given that the input video sequence is monocular and the scene is not static? Can you share more details about about the (frozen) initialization of the forward / backward deformation networks when training the canonical Gaussians and the SfM initialization?
- Are there any qualitative observations relating to decomposition of geometry / appearance information between the mesh representation and the gaussians?
- How do you arrive at running mesh anchoring every 100 iteration? Is it possible to perform differentiable mesh anchoring?

---

> ### Author Response · Authors · 2024-11-22
> **Response to Reviewer 44mR (1/2)**
>
> We thank the reviewer for their valuable feedback. We address your comments in the following.
>
> ---
>
> **Q:** *How / if appearance improves between the baselines and the proposed method when the underlying geometry is of similar quality?*
>
> **A:** Following the reviewer's suggestion, we use the geometry generated by DG-Mesh and render the mesh using appearance queried from other baseline methods. The results below demonstrate that, **even when the underlying geometry quality is comparable, DG-Mesh consistently achieves better visual quality compared to other approaches**. The surface rendering approach in DG-Mesh enables it to learn appearance directly and precisely at the object's surface. In contrast, other volume rendering-based methods fail to store appearance details accurately at the surface, leading to less precise results.
>
> | Methods | PSNR_m ↑ | SSIM ↑ | Novel View Rendering (Images) |
> |:----------|:---:|:---:|:---:|
> | D-NeRF | 24.169 | 0.905 | [#1](https://dgmesh.github.io/rebuttal_materials/fix_geo_images/dnerf/dnerf-jumpingjacks.png), [#2](https://dgmesh.github.io/rebuttal_materials/fix_geo_images/dnerf/dnerf-hook.png), [#3](https://dgmesh.github.io/rebuttal_materials/fix_geo_images/dnerf/dgmesh-horse.png), [#4](https://dgmesh.github.io/rebuttal_materials/fix_geo_images/dnerf/dgmesh-bird.png) |
> | HexPlane | 29.013 | 0.954 | [#1](https://dgmesh.github.io/rebuttal_materials/fix_geo_images/hexplane/dnerf-jumpingjacks.png), [#2](https://dgmesh.github.io/rebuttal_materials/fix_geo_images/hexplane/dnerf-hook.png), [#3](https://dgmesh.github.io/rebuttal_materials/fix_geo_images/hexplane/dgmesh-horse.png), [#4](https://dgmesh.github.io/rebuttal_materials/fix_geo_images/hexplane/dgmesh-bird.png) |
> | TiNeuVox-B | 28.157 | 0.943 | [#1](https://dgmesh.github.io/rebuttal_materials/fix_geo_images/tineuvox/dnerf-jumpingjacks.png), [#2](https://dgmesh.github.io/rebuttal_materials/fix_geo_images/tineuvox/dnerf-hook.png), [#3](https://dgmesh.github.io/rebuttal_materials/fix_geo_images/tineuvox/dgmesh-horse.png), [#4](https://dgmesh.github.io/rebuttal_materials/fix_geo_images/tineuvox/dgmesh-bird.png) |
> | **DG-Mesh (Ours)** | **29.790** | **0.959** | [#1](https://dgmesh.github.io/rebuttal_materials/fix_geo_images/dgmesh/dnerf-jumpingjacks.png), [#2](https://dgmesh.github.io/rebuttal_materials/fix_geo_images/dgmesh/dnerf-hook.png), [#3](https://dgmesh.github.io/rebuttal_materials/fix_geo_images/dgmesh/dgmesh-horse.png), [#4](https://dgmesh.github.io/rebuttal_materials/fix_geo_images/dgmesh/dgmesh-bird.png) |
>
> ---
>
> **Q:** *Lack of rigorous quantitative evaluation for several key decisions such as choosing the initialization, DMTet/DiffMC/FlexiCubes, mesh anchoring frequency, and the training methodology.*
>
> **A:**
> * Choosing the initialization and training methodology
>     * Our choice of initialization follows established practices in prior works, including 3DGS [1] and Deformable-3DGS [2], which initialize the Gaussians from SfM points and adopt a warm-up stage where the canonical Gaussians were optimized without deforming.
> * DMTet/Diff.M.C./FlexiCubes
>     * Directly replacing Diff.M.C. with DMTet or FlexiCubes in our pipeline is infeasible because both DMTet and FlexiCubes operate by querying their grid values from a continuous SDF space. However, there is currently no method available to transform oriented points into a continuous SDF function in a differentiable manner without GT shape supervision. To handle this limitation, DG-Mesh converts the oriented Gaussian points into a discretized SDF value grid using DPSR and then applies Diff.M.C. to extract the surface from the value grid.
> * Mesh anchoring frequency
>     * We evaluate the impact of different anchor intervals and provide their quantitative results below. The mesh geometry quality deteriorates as the anchor intervals increase since larger intervals result in fewer anchoring steps, reducing the alignment between the Gaussians and the surface.
>
>     | Anchor Interval | PSNR_m ↑ | CD(10⁻²) ↓ | EMD(10⁻¹) ↓ |
>     |:---:|:---:|:---:|:---:|
>     | 50 | 30.521 | 0.314 | 1.307 |
>     | 100 | **30.668** | **0.306** | 1.298 |
>     | 200 | 30.599 | 0.319 | **1.295** |
>     | 500 | 30.645 | 0.320 | 1.298 |

---

> ### Author Response · Authors · 2024-11-22
> **Response to Reviewer 44mR (2/2)**
>
> **Q:** *What's the intuition behind the improvement when using both Gaussian and mesh rendering losses?*
>
> **A:** Gaussian rendering provides direct guidance in learning the dynamics of Gaussian points, while mesh rendering ensures that the surface extracted from the oriented Gaussian points is accurate in both geometry and appearance. By combining these two rendering losses, we ensure that the reconstructed mesh sequence is both geometrically precise and physically realistic.
>
> ---
>
> **Q:** *Performance on a large synthetic textured but planar surface where there might not be enough vertices to anchor gaussians to?*
>
> **A:** The number of our mesh vertices is determined by the grid resolution during iso-surface extraction. Therefore, a planar surface can still be described by a sufficient number of vertices for Gaussians to anchor to. To fully address the reviewer's concern, we constructed a synthetic dataset featuring a large textured board with planar surfaces (sample data provided here: [[Images]](https://dgmesh.github.io/rebuttal_materials/planar_data.png)). The quantitative results below, along with novel view renderings, demonstrate that DG-Mesh performs effectively on large planar surfaces, maintaining high fidelity and consistency in both surface reconstruction and appearance rendering.
>
> | PSNR_m ↑ | SSIM ↑ | CD ↓ | EMD ↓ | Novel View and Mesh |
> |:---:|:---:|:---:|:---:|:---:|
> | 23.264 | 0.828 | 0.489 | 1.528 | [[Video]](https://dgmesh.github.io/rebuttal_materials/planar_video.mp4) |
>
> ---
>
> **Q:** *What do blue dots in anchored Gaussians represent?*
>
> **A:** They represent the Gaussian's mean position after anchoring. We will add annotations in the revised version to ensure better understanding for the readers.
>
> ---
>
> **Q:** *How is the "canonical" frame chosen? More details about the (frozen) initialization of the forward / backward deformation networks when training the canonical Gaussians and the SfM initialization?*
>
> **A:** There is no "canonical frame.” Instead, we maintain a **“canonical space” that does not correspond to any specific frame from the input video**. During the initialization phase, the deformation networks are frozen, and the canonical Gaussians take SfM points as their initial positions and are optimized without applying any deformation. Since the canonical Gaussians are supervised with dynamic inputs, they capture a unified geometric structure that integrates geometric information across all time frames, which provides a robust initialization for the deformation networks in subsequent training stages.
>
> ---
>
> **Q:** *Qualitative observations relating to decomposition of geometry / appearance information between the mesh representation and the gaussians?*
>
> **A:** We visualize the reconstructed surface geometry, mesh appearance, Gaussian geometry, and Gaussian appearance in videos ([#1](https://dgmesh.github.io/rebuttal_materials/decomposition/jumpingjacks.mp4), [#2](https://dgmesh.github.io/rebuttal_materials/decomposition/hook.mp4), [#3](https://dgmesh.github.io/rebuttal_materials/decomposition/mutant.mp4)).
>
> ---
>
> **Q:** *How do you arrive at running mesh anchoring every 100 iterations? Is it possible to perform differentiable mesh anchoring?*
>
> **A:** We set the anchoring interval to 100, as it represents a balanced trade-off between maintaining geometry quality and achieving optimal rendering performance. Please refer to our response to the second question where we evaluate the impact of different anchor intervals and provide their quantitative results. Our anchoring process includes a differentiable anchoring loss and a non-differentiable deletion/densification process. The anchoring loss provides the gradient to align the Gaussians toward the mesh surfaces.
>
> ---
>
> Please do not hesitate to let us know if you have any additional comments.
>
> **References**
>
> [1] *Kerbl, Bernhard, et al. "3D Gaussian Splatting for Real-Time Radiance Field Rendering." TOG, 2023.*
>
> [2] *Yang, Ziyi, et al. "Deformable 3d gaussians for high-fidelity monocular dynamic scene reconstruction." CVPR, 2024.*

---

> ### Author Response · Authors · 2024-11-24
> **Response to Reviewer 44mR**
>
> Dear Reviewer 44mR,
>
> Thank you again for your detailed feedback. As we approach the end of the author-reviewer discussion period, we notice that there have not yet been any responses to our rebuttal.
>
> Please feel free to request any additional information or clarification that may be needed. We hope to deliver all the information in time before the deadline.
>
> Thank you!

---

> > ### Comment · Reviewer_44mR · 2024-11-26
> >
> > Thank you for the thorough rebuttal - these have adequately addressed all my concerns. I've updated my rating for this submission.

---

> ### Author Response · Authors · 2024-11-26
> **Response to Reviewer 44mR**
>
> Dear Reviewer 44mR,
>
> Thank you for taking the time to review our responses. And we are glad that the questions have been resolved! These experiments and discussions have been included in our revised paper and supplement.

---

### Comment · Area_Chair_MAAg · 2024-11-21
**Please initiate discussions!**

Dear authors and reviewers,

The discussion phase has already started. You are highly encouraged to engage in interactive discussions (instead of a single-sided rebuttal) before November 26. Please exchange your thoughts on the submission and reviews at your earliest convenience.

Thank you,
ICLR 2025 AC

---

### Author Response · Authors · 2024-11-22
**General Response (1/2)**

We appreciate the reviewers for acknowledging the novelty of our approach (Reviewer `44mR`, `wAGq`, `9ERR`) and recognizing the high quality of our mesh reconstruction (Reviewer `44mR`, `wAGq`, `9ERR`). Additionally, we thank Reviewer `9ERR`for highlighting that our work is pioneering in this field.

We would like to emphasize the main focus of this paper is to **reconstruct a consistent mesh sequence from dynamic input, achieving both a topologically accurate surface and a consistent appearance.** While recent dynamic Gaussian works have sought to enhance the consistency of Gaussian motion and visual quality using depth information or physical priors, **none of them** deliver surface sequences that are consistent in topology and appearance. In our approach, the Gaussian-Mesh Anchoring establishes a robust mapping between the Gaussian distribution and the mesh's topology for each time frame. By combining this with the Cycle-Consistent Deformation on Gaussians, we ensure a coherent mesh sequence, where mesh faces maintain consistent correspondence throughout the sequence. The effectiveness of our approach is evident in the results, as we have successfully constructed high-quality, time-consistent meshes using everyday videos captured with simple handheld phones. **To the best of our knowledge, no prior work has accomplished this.**

Below, we address the reviewers' feedback by providing baseline comparisons with additional recent dynamic Gaussian methods and clarifying details on the color implementation of DG-Mesh.

---

> ### Author Response · Authors · 2024-11-22
> **General Response (2/2)**
>
> ### (1) Comparison with additional recent baselines. [Reviewer `mkQh`, `9ERR`]
>
> We have extended our baseline comparisons to include additional recent dynamic Gaussian methods [1, 2, 3], as suggested. Specifically, we compare our approach with SC-GS [1] and 4D-GS [2], providing their results below. Notably, Spacetime-GS [3] directly fails at the D-NeRF and DG-Mesh datasets since it assumes multi-view inputs. DG-Mesh consistently outperforms these baselines and achieves the best geometric quality.
>
>
> | | CD (10⁻²) ↓ | EMD (10⁻¹) ↓ | Mesh Sequence (Videos) |
> |:---:|:---:|:---:|:---:|
> | SC-GS | 0.458 | 1.691 | [#1](https://dgmesh.github.io/rebuttal_materials/baseline_videos/scgs_horse.mp4), [#2](https://dgmesh.github.io/rebuttal_materials/baseline_videos/scgs_bird.mp4), [#3](https://dgmesh.github.io/rebuttal_materials/baseline_videos/scgs_girlwalk.mp4), [#4](https://dgmesh.github.io/rebuttal_materials/baseline_videos/scgs_beagle.mp4) |
> | 4DGS | 0.974 | 2.405 | [#1](https://dgmesh.github.io/rebuttal_materials/baseline_videos/4dgs_horse.mp4), [#2](https://dgmesh.github.io/rebuttal_materials/baseline_videos/4dgs_bird.mp4), [#3](https://dgmesh.github.io/rebuttal_materials/baseline_videos/4dgs_girlwalk.mp4), [#4](https://dgmesh.github.io/rebuttal_materials/baseline_videos/4dgs_beagle.mp4) |
> | **DG-Mesh (Ours)** | **0.306** | **1.298** | [#1](https://dgmesh.github.io/rebuttal_materials/baseline_videos/dgmesh_horse.mp4), [#2](https://dgmesh.github.io/rebuttal_materials/baseline_videos/dgmesh_bird.mp4), [#3](https://dgmesh.github.io/rebuttal_materials/baseline_videos/dgmesh_girlwalk.mp4), [#4](https://dgmesh.github.io/rebuttal_materials/baseline_videos/dgmesh_beagle.mp4) |
>
> ### (2) How is the color of the mesh predicted? [Reviewer `wAGq`, `mkQh`]
>
> DG-Mesh handles the appearance in its two rasterization passes differently. For the Gaussian rasterizer, the color is derived from the volume rendering results of the spherical harmonics (SH) stored on each Gaussian. For the mesh rasterizer, the color is obtained from a learned appearance MLP that predicts time-dependent surface color. To query the surface color of a deformed mesh for each time frame, the vertex positions are projected back into the canonical space, where the appearance MLP is queried to generate the corresponding vertex colors. We observe that learning the appearance in the canonical space improves appearance consistency across time frames, resulting in a more coherent and stable visual representation.
>
> ---
>
> **References**
>
> [1] *Huang, Yi-Hua, et al. "Sc-gs: Sparse-controlled gaussian splatting for editable dynamic scenes." CVPR, 2024.*
>
> [2] *Yang, Zeyu, et al. "Real-time photorealistic dynamic scene representation and rendering with 4d gaussian splatting." ICLR, 2024.*
>
> [3] *Li, Zhan, et al. "Spacetime gaussian feature splatting for real-time dynamic view synthesis." CVPR, 2024.*

---

### Comment · Area_Chair_MAAg · 2024-11-25
**Last day for interactive discussions!**

Dear authors and reviewers,

The interactive discussion phase will end in one day (November 26). Please read the authors' responses and the reviewers' feedback carefully and exchange your thoughts at your earliest convenience. This would be your last chance to be able to clarify any potential confusion.

Thank you,
ICLR 2025 AC

---

### Author Response · Authors · 2024-11-26
**Paper Revision Updates**

We have submitted a revised version of our paper based on the reviewers' feedback. The changes are highlighted in blue. Below are the key updates:

- Revised the paper's title to better reflect the scope of our contributions.
- Added comparison results with 4DGS and SC-GS.
- Added new experiments on the monocular Dycheck dataset and highlighted multiview results on the NeuralActor dataset.
- Added ablation studies, including experiments on the appearance module and anchoring frequency.

Again, we sincerely thank all the reviewers for their constructive feedback.

---

### Meta-Review · Area_Chair_MAAg · 2024-12-19

**Metareview:**

The submission received positive reviews from all the reviewers. The reviewers generally appreciate the presentation, recognize the novelty of the method, and are convinced by the positive experimental results. After reading the paper, the reviewers' comments and the authors' rebuttal, the AC agrees with the decision by the reviewers and recommends acceptance.

**Additional Comments On Reviewer Discussion:**

The reviewers raised several questions regarding lack of quantitative evaluation (44mR), clarifications on method details (wAGq, 9ERR), and evaluation (44mR, wAGq). The questions were addressed by the authors in good detail. Reviewers 44mR, wAGq, and 9ERR were convinced by the responses and raised their ratings. The AC agrees with the evaluation.

---

### Decision · Program_Chairs · 2025-01-22

Accept (Poster)